# DreamOmni2: Multimodal Instruction-based Editing and Generation

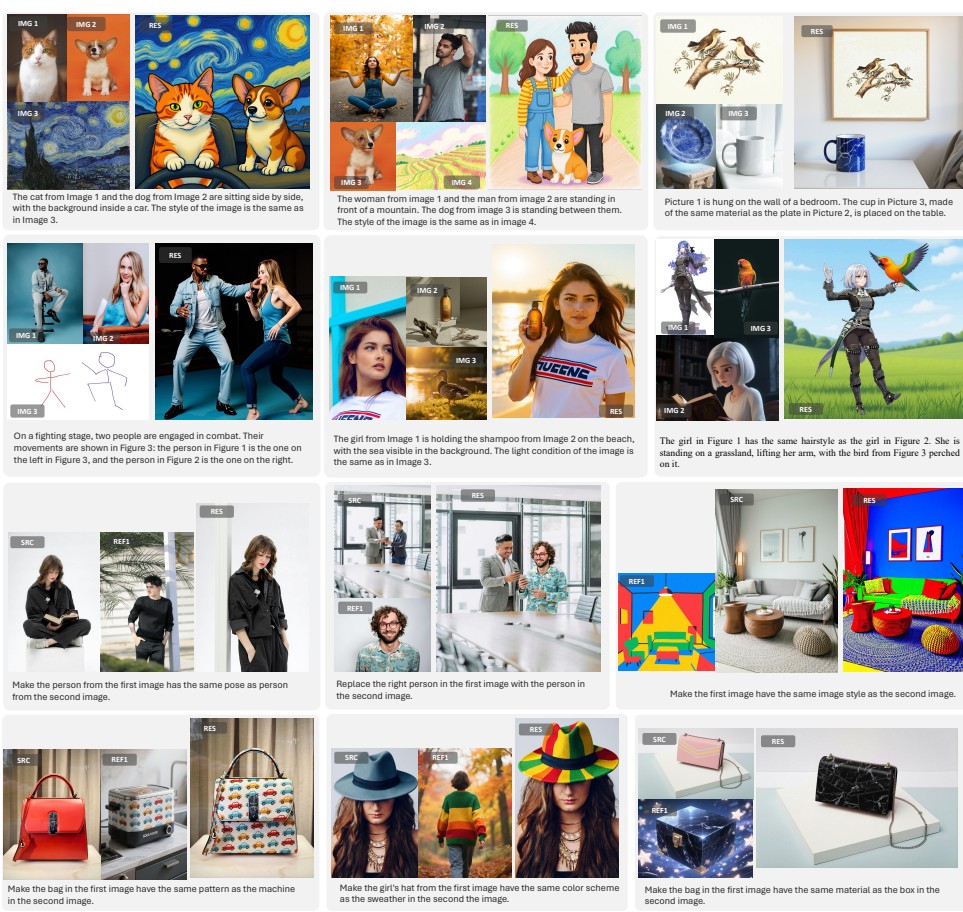

Figure 1: The gallery of DreamOmni2: Enabling multimodal instruction-based editing and generation, extending beyond concrete objects to abstract attributions.

## ABSTRACT

Recent advancements in instruction-based image editing and subject-driven generation have garnered significant attention, yet both tasks still face limitations in meeting practical user needs. Instruction-based editing relies solely on language instructions, which often fail to capture specific editing details, making reference images necessary. Meanwhile, subject-driven generation is limited to combining concrete objects or people, overlooking broader, abstract concepts. To address these challenges, we propose two novel tasks: multimodal instruction-based editing and generation. These tasks support both text and image instructions and extend the scope to include both concrete and abstract concepts, greatly enhancing their practical applications. We introduce DreamOmni2, tackling two primary challenges: data creation and model framework design. Our data synthesis pipeline consists of three steps: (1) using a feature mixing method to create extraction data for both abstract and concrete concepts, (2) generating multimodal instruction-based editing training data using the editing and extraction models,

and (3) further applying the extraction model to create training data for multimodal instruction-based editing. For the framework, to handle multi-image input, we propose an index encoding and position encoding shift scheme, which helps the model distinguish images and avoid pixel confusion. Additionally, we introduce joint training with the VLM and our generation/editing model to better process complex instructions. In addition, we have proposed comprehensive benchmarks for these two new tasks to drive their development. Experiments show that DreamOmni2 has achieved impressive results. Models and codes will be released.

# 1 INTRODUCTION

Recent advancements in unified generation and editing models (OpenAI, 2025; Google, 2025b) have gained significant attention and praise in the market. The success of these models can be attributed to several factors: **(1)** They greatly improve user experience by simplifying the process, allowing users to perform various design tasks within a single model without the need to switch between different ones. **(2)** Unified models reduce deployment costs for service providers. **(3)** Academically, they contribute to the exploration of AGI and world models, enabling the accurate understanding of user instructions and the creation or modification of real-world visual content.

Current released works (Batifol et al., 2025; Wu et al., 2025a; Deng et al., 2025) mainly focus on instruction-based editing and subject-driven generation with a text prompt and a single source image input, but both have limitations in application and advancing intelligence. **(1)** For instruction-based editing (Brooks et al., 2023; Liu et al., 2025; Xia et al., 2025b), instructions alone often fail to fully capture the user's intent. For example, when a user says, "make the bag in the image have the same pattern as the dress in the given image," it's difficult to describe the complex pattern of "dress" with words. Thus, accurate editing requires multimodal instructions, including reference images and text. Notably, this challenge involves not only modifying objects but also any abstract attributes, such as texture, material, posture, hairstyle, and design style, which are difficult to describe with words. **(2)** Subject-driven generation models (Xiao et al., 2025; Wu et al., 2025c) and even commercial unified models (Google, 2025b) mainly focus on generating content from specific concrete objects or people, with limited research on referencing more general abstract attributions from input images.

To create a more intelligent and all-encompassing unified creation tool, we propose DreamOmni2. The biggest challenge lies in the training data, so we introduce a comprehensive data pipeline for multimodal instruction-based editing and generation, consisting of the following steps (Fig. 2): **(1)** We propose a feature mixing scheme to exchange attention features between two batches, allowing the model to generate pairs of images with the same abstract attribute or concrete object. Compared to the previous diptych method (Wu et al., 2025c) for generating image pairs, our scheme achieves a higher success rate, produces images with greater resolution, and completely eliminates any content blending at the edges when the pair of images is split. **(2)** Using the pairs generated in Step 1, we train a generation-editing model as an extraction model. This model extracts concrete objects or abstract attributions from the given image and generates another based on instructions. Compared to previous methods (Wu et al., 2025c; Chen et al., 2025) relying on segmentation and detection, our extraction model offers three key advantages: it can handle abstract concepts, occluded objects, and generate more diverse reference images. We then generate multimodal instruction-based editing training data, which includes a target image, a source image, an editing instruction, and multiple reference images. We use a text-to-image (T2I) model to generate a target image based on multiple keywords or select one from a real image database. The extraction model then generates reference images for one of the keywords. Additionally, we use an instruction-based editing model (Batifol et al., 2025) to transform the content defined by the selected keyword into something different, obtaining the source image. **(3)** We create multimodal instruction-based generation data by applying the extraction model to generate several reference images based on keywords from the source images created in Step 2. Thus, we build data for generating images from multiple reference images.

Furthermore, the current SOTA unified generation and editing models (Batifol et al., 2025) still cannot handle multiple image inputs. To this end, we propose the Dreamomni2 framework. First, we propose an index encoding and position encoding shift scheme. Index encoding helps the model identify the input image's index, improving its understanding of the referenced image in the instructions. Position encoding is shifted based on previous inputs, preventing pixel confusion and the copy-and-paste effect in the generated results. In addition, we propose a joint training scheme

for the generation/editing model and VLM. While instructions in generation and editing models are typically simple, real-world user instructions are often irregular and logically complex. A VLM pre-trained on large-scale corpora can better understand these complex intentions, translating instructions into a form the model can comprehend, significantly improving performance in real-world scenarios. Furthermore, for these two new tasks, we have built the DreamOmni2 benchmark using real image data. This allows for a more accurate assessment of the model's generalization and performance in real-world scenarios. Our main contributions are fourfold:

- We propose two highly practical tasks: multimodal instruction-based editing and generation guided by any concrete or abstract concept. Introducing these two tasks makes current unified generation and editing models smarter and more versatile creative tools.
- We propose a three-stage data creation pipeline. Leveraging this pipeline, we have built a high-quality, comprehensive multimodal instruction-based editing and generation dataset.
- We present the DreamOmni2 framework, which introduces the index encoding and position encoding shift scheme, enabling the model to handle multi-reference image inputs. Additionally, we propose a joint training scheme for the generation/editing model and VLM, enhancing the model's ability to understand complex user instructions.
- For these two new tasks, we propose a DreamOmni2 benchmark built from real image data. Experiments demonstrate the effectiveness of DreamOmni2 in real-world scenarios.

## 2 RELATED WORK

**Instruction-based Editing** refers to modifying an image based on a user's language instruction (Deng et al., 2025; Sheynin et al., 2023; Xia et al., 2024). The main challenge of this task lies in the creation of high-quality and accurate editing datasets. As a pioneering work, InstructP2P (Brooks et al., 2023) introduced an instruction-based image editing dataset by fine-tuning GPT-3 and Prompt-to-Prompt (Hertz et al., 2022) with SD 1.5 (Rombach et al., 2022). Since then, many other approaches (Zhang et al., 2024; Xia et al., 2025b; Wei et al., 2024; Ge et al., 2024) for creating datasets have emerged, such as employing people to create data, using inpainting methods, collage-based methods, and using different expert models. Recently, DreamVE (Xia et al., 2025a) has unified instruction-based image and video editing. However, language-based editing is limited, as many details in real-world scenarios can't be captured with words, requiring reference images for better description. To this end, we propose multimodal instruction-based editing, enabling guidance from concrete objects or abstract attributes in reference images. This makes unified image generation and editing models (Batifol et al., 2025; Wu et al., 2025a) more comprehensive and practical.

**Subject-driven Generation** has been extensively studied. Methods like Dreambooth (Ruiz et al., 2023) and textual inversion (Gal et al., 2022) fine-tune models on multiple images of the same subject, enabling subject-driven generation. However, this requires users to prepare several images and perform fine-tuning for each new subject, which is not user-friendly. Later approaches like IP-adapter (Ye et al., 2023) and BLIP-diffusion (Li et al., 2023) used visual encoders to compress the subject of a reference image into a vector and inject it into a diffusion model, enabling subject-driven generation without fine-tuning. IC LoRA (Huang et al., 2024) and Ominicontrol (Tan et al., 2024) further explored the inherent image reference capabilities of DIT models (Peebles & Xie, 2023). Recently, unified generation and editing models (Xia et al., 2025b; Batifol et al., 2025; Wu et al., 2025a; Xiao et al., 2025) have adopted the simple approach of encoding reference images as visual tokens, concatenating them with text and noise tokens, and feeding them into the DIT model. However, prior methods focus mainly on concrete objects, limiting their ability to capture broader abstract concepts. In this paper, we propose multimodal instruction-based generation, a task that enables referencing any concrete objects or abstract attributes in reference images to generate new ones. This extends the scope of subject-driven generation and enhances its practicality.

## 3 METHODOLOGY

### 3.1 SYNTHETIC DATA

Multimodal instruction-based editing and generation are new tasks, with the main challenge being the lack of training data. For multimodal instruction-based editing, the previous data creation pipeline (Brooks et al., 2023; Wei et al., 2024) involves generating triplets of instructions, source

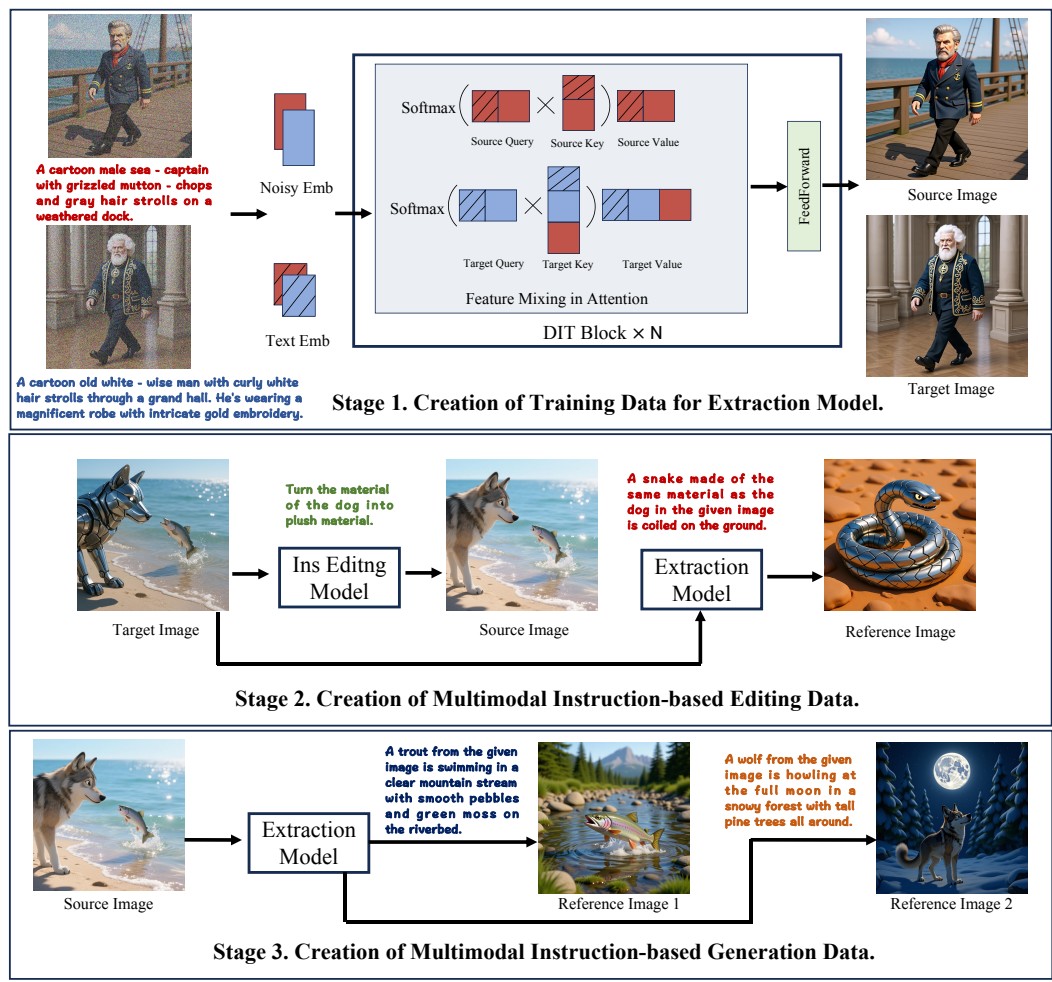

Figure 2: The Overview of DreamOmni2's training data construction. **(1)** In stage 1, we use a feature mixing scheme to leverage the base model's T2I capabilities, creating high-quality data pairs with concrete objects and abstract attributes. **(2)** In stage 2, we generate multimodal instruction-based editing data. Using stage 1 data, we train an extraction model to simulate objects or attributes in the target image and generate a reference image based on instructions. Additionally, we use an instruction-based editing model to modify the extracted objects or attributes in the target image to be different, creating the source image. This generates training pairs from reference and source images to the target image. **(3)** In stage 3, we extract objects from stage 2's source images to create new reference images, forming training data for generating target images from reference images.

images, and target images. However, this approach does not allow for creating data that incorporates reference images as a condition for editing. For multimodal instruction-based generation, the previous subject-generation data pipeline (Wu et al., 2025c; Chen et al., 2025) relies on segmentation detection models to create reference images. This approach makes it difficult to synthesize data for generating reference abstract attributions or occluded concrete objects.

To address the training data problem for these two tasks, we propose a comprehensive synthetic data pipeline. Specifically, as illustrated in Fig. 2, our approach consists of three stages. In the first stage, we introduce a feature mixing scheme, where a dual-branch structure is employed to simultaneously generate both the source image and the target image as follows:

$$\text{Attn}_{tar}(\boldsymbol{Q}, \boldsymbol{K}, \boldsymbol{V}) = \text{softmax}\left(\frac{\boldsymbol{Q}\boldsymbol{K}^{\top}}{\sqrt{d}}\right)\boldsymbol{V}, \tag{1}$$

where $Q = [Q^n_{tar}; Q^t_{tar}]$, $K = [K^n_{tar}; K^t_{tar}; K^n_{src}]$, and $V = [V^n_{tar}; V^t_{tar}; V^n_{src}]$. $Q^t_{tar}$, $K^t_{tar}$, and $V^t_{tar}$ are the text features from the target branch, while $Q^n_{tar}$, $K^n_{tar}$, and $V^n_{tar}$ are the noise features

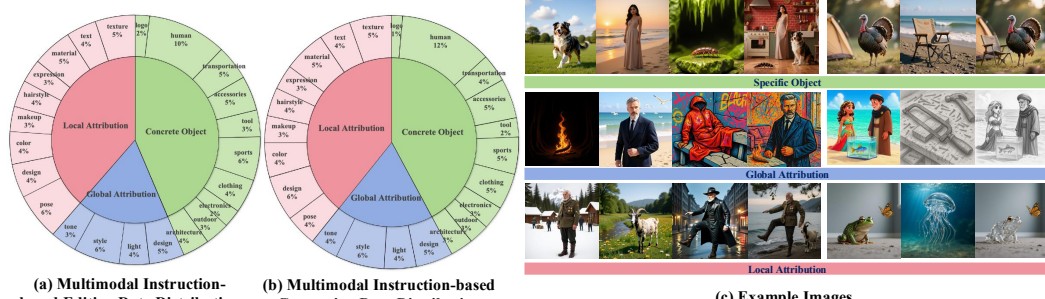

**(a) Multimodal Instruction-based Editing Data Distribution**

**(b) Multimodal Instruction-based Generation Data Distribution**

**(c) Example Images**

Figure 3: Data distribution and samples for multimodal instruction-based editing and generation training data. Our dataset is comprehensive and diverse, including the generation and editing of concrete objects as well as abstract attributions, such as local and global attributions.

from the target branch. $K_{src}^n$ and $V_{src}^n$ are the noise features from the source branch at the same layer as the $K_{tar}^n$ and $V_{tar}^n$. $[;]$ indicates token (or called length) dimension concatenation.

Our feature mixing scheme leverages the model's inherent T2I capability to generate paired training data. Compared to the previous UNO (Wu et al., 2025c) diptych generation method, our feature mixing scheme has several clear advantages: **(1)** The diptych method halves the image resolution by forcing two images into one, while Feature Mixing generates in two branches without reducing resolution. **(2)** The diptych approach often misplaces the dividing line, leading to content blending. Our method avoids this issue. **(3)** Data generated by the feature mixing scheme is of higher quality and accuracy than that from the diptych approach. Then, we use the data to train extraction models. Our training data not only enhances the base model (Batifol et al., 2025)'s ability to extract concrete objects but also enables it to capture abstract concepts, a capability it previously lacked.

Afterward, as shown in Fig. 2 stage 2, we create multimodal instruction-based editing data. Specifically, we first create target images, using both T2I model-generated data and real images. For T2I-generated images, we randomly select diverse element keywords (e.g., objects or attributes) and use an LLM to compose a prompt, which the T2I model then uses to generate the target image. For real images, we directly use a VLM to extract keywords. T2I data is more flexible, allowing any concept combination, while real images reflect natural distributions. Thus, we combine both types of data. Next, using the extraction model trained in stage 1, we extract an object or attribution from the target image based on a selected keyword to create a reference image. We then apply instruction-based editing model (Batifol et al., 2025) to alter the selected keyword in the target image, obtaining the source image. Finally, we use an LLM to generate the editing instructions, forming a training tuple consisting of the source image, instruction, reference image, and target image.

After that, as shown in Fig. 2 stage 3, we create multimodal instruction-based generation data. We use the extraction model to extract keywords from the source image in stage 2, generating reference images. By combining these with the reference images from stage 2, we can obtain training tuples consisting of multiple reference images, an instruction, and a target image.

Our created dataset is shown in Fig. 3. Our dataset includes both real and synthetic target data, covering a wide range of object categories for generation and editing, including various abstract attributions and concrete objects. Additionally, we provide a comprehensive set of reference images, with cases ranging from one to five references, enabling the model to handle a wide variety of tasks.

### 3.2 FRAMEWORK AND TRAINING

The unified generation and editing base model (Batifol et al., 2025) can only process a single input image. To this end, we propose the DreamOmni2 framework. In multimodal instruction-based tasks, users typically reference images as "image 1", "image 2" for convenience. However, in DIT, positional encoding alone cannot accurately distinguish the index of reference images. Therefore, we solve this by adding an index encoding to positional channels. Although index encoding helps distinguish reference images, we found that the position encoding still requires an offset based on the size of the previously input reference images. By adding this offset to the position encoding, we observed a reduction in copy-and-paste artifacts and pixel confusion between reference images.

Table 1: Comparison between our DreamOmni2 benchmark and existing related benchmarks.

| Benchmarks | Task Type | Num Reference | Editing Target Concrete Object | Abstract Attribution |
|---|---|---|---|---|
| DreamBooth (Ruiz et al., 2023) | Generation | Single | ✓ | ✗ |
| OmniContext (Wu et al., 2025b) | Generation | Multiple | ✓ | ✗ |
| DreamOmni2 (Ours) | Generation & Editing | Multiple | ✓ | ✓ |

Currently, training instructions for generation and editing models are usually well-structured with a fixed format. However, real-world user instructions are often irregular or logically inconsistent, creating a gap that can hinder the model's understanding and reduce performance. To address this, we propose joint training of the VLM and generation models, enabling the VLM to interpret complex user instructions and output them in the structured format used in training, helping the editing and generation model better understand user intent. For multimodal instruction-based editing, the predefined output format combines user instructions with refined image descriptions, while for multimodal instruction-based generation, the VLM directly outputs a refined image description.

During training, we fine-tune Qwen2.5-VL (Wang et al., 2024) 7B to learn the predefined standard output format, with a learning rate of $1 \times 10^{-5}$, using approximately 10 A100 hours. We then train the editing and generation models using LoRA on Flux Kontext (Batifol et al., 2025) to perform multimodal instruction-based editing and generation with the predefined standard instruction format. Notably, by using LoRA for training, we can retain the original instruction-editing capabilities of Kontext. As soon as a reference image is detected, our LoRA is activated, seamlessly integrating multimodal instruction-based editing and generation into the unified model. Additionally, we train LoRA for generation and editing separately, as the distinction between generation and editing lies in whether the consistency of the source image is preserved. Since instructions often do not clarify whether the user intends to edit or generate, separate training allows users to make their own choice. Both DreamOmni2 editing and generation LoRA are trained on a batch size of 16 and a learning rate of $5 \times 10^{-6}$, consuming about 384 A100 hours.

### 3.3 BENCHMARK

Currently, no benchmark exists for multimodal instruction-based editing and generation. As shown in Tab. 1, DreamBooth (Ruiz et al., 2023) only supports single-image generation. Although Omni-Context (Wu et al., 2025b) includes some multi-reference testing cases, it focuses solely on concrete object combinations and does not evaluate multimodal instruction-based editing or the inclusion of abstract attributes. To address this, we propose the DreamOmni2 benchmark to drive progress in these areas. Our benchmark is comprehensive, consisting of real images to accurately assess the model's performance in real-world scenarios. The test cases cover a variety of categories, including the reference generation and editing of abstract attributions (global and local) and concrete objects. More details about our DreamOmni2 benchmark can be found in the Appendix.

## 4 EXPERIMENTS

**Evaluation on Multimodal Instruction-based Image Editing.** As shown in Tab. 2, we compare several competitive models that natively support multiple image inputs, such as DreamO (Mou et al., 2025), Omnigen2 (Wu et al., 2025b), and Qwen-image-Edit-2509 (Wu et al., 2025a). Although Kontext (Batifol et al., 2025) and Qwen-image-Edit (Wu et al., 2025a) do not natively support multiple image inputs, we applied the method from Diffusers (von Platen et al., 2022), which combines multiple images into one input. We also compared closed-source commercial models, such as Nano Banana (Google, 2025b) and GPT-4o (OpenAI, 2025). We tested editing examples with concrete objects and abstract attributions on DreamOmni2 benchmark. The models were evaluated for success rates by Gemini 2.5 (Google, 2025a) and Doubao 1.6 (ByteDance, 2025), and several professional engineers manually assessed the results. As shown in Tab. 2, our DreamOmni2 achieved the best performance in human evaluations. In VLM tests, DreamOmni2 significantly outperformed open-source models and achieved results close to those of commercial models. In fact, GPT-4o and Nano Banana often introduced unintended changes or inconsistencies in the edited attribution, which were not aligned with the reference images. These issues are difficult for VLMs to detect accurately. Additionally, GPT-4o caused the edited images to appear yellowed.

Qualitative results are shown in Fig. 4, where we present visualizations of editing cases involving various concrete objects and abstract attributes. It is clear that DreamOmni2 produces more accurate

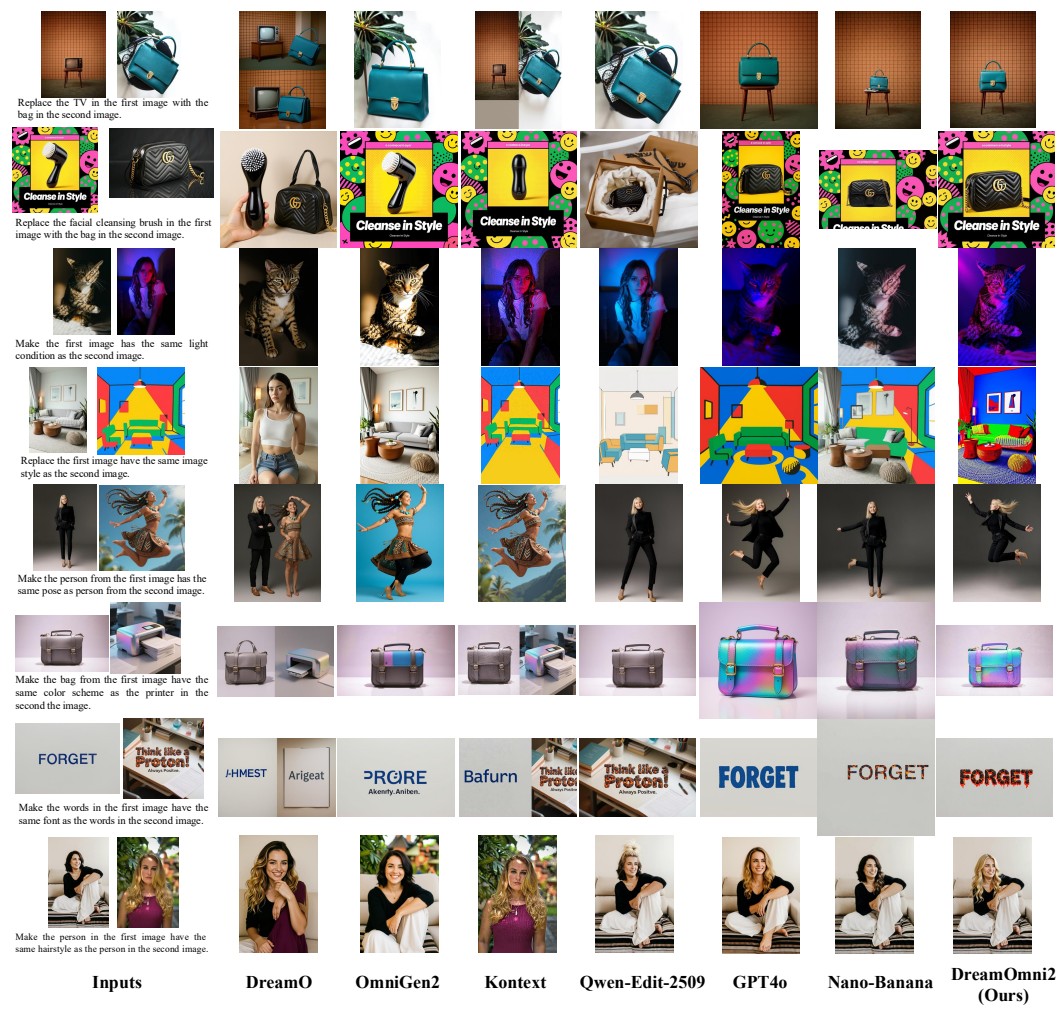

Figure 4: Visual comparison of multimodal instruction-based editing. Compared to other competitive methods and even closed-source commercial models (GPT-4o and Nano Banana), DreamOmni2 shows more accurate editing results and better consistency.

Table 2: Quantitative comparison of multimodal instruction-based editing. We use Gemini (Google, 2025a) and Doubao (ByteDance, 2025) to evaluate the success editing ratio of different models on concrete objects and abstract attributions, respectively. In addition, "Human" refers to professional engineers assessing the editing success rates of all models.

| Method | Concrete Object | | | Abstract Attribution | | |
|---|---|---|---|---|---|---|
| | Gemini↑ | Doubao↑ | Human↑ | Gemini↑ | Doubao↑ | Human↑ |
| GPT-4o (OpenAI, 2025) | 0.6829 | 0.7805 | 0.5610 | 0.7195 | 0.7439 | 0.5793 |
| Nano Banana (Google, 2025b) | 0.6829 | 0.7073 | 0.5366 | 0.6463 | 0.5488 | 0.3293 |
| UNO (Wu et al., 2025c) | 0.0000 | 0.0244 | 0.0000 | 0.0061 | 0.0183 | 0.0000 |
| DreamO (Mou et al., 2025) | 0.0244 | 0.0732 | 0.0000 | 0.0183 | 0.0183 | 0.0000 |
| Omnigen2 (Wu et al., 2025b) | 0.2195 | 0.2927 | 0.2927 | 0.0427 | 0.0793 | 0.0305 |
| Qwen-Image-Edit (Wu et al., 2025a) | 0.0976 | 0.1463 | 0.0244 | 0.0244 | 0.0183 | 0.0000 |
| Kontext (Batifol et al., 2025) | 0.0488 | 0.1220 | 0.0976 | 0.0183 | 0.0122 | 0.0122 |
| Qwen-Image-Edit-2509 (Wu et al., 2025a) | 0.2683 | 0.2927 | 0.2195 | 0.0488 | 0.1159 | 0.0427 |
| DreamOmni2 (Ours) | **0.5854** | **0.6585** | **0.6098** | **0.5854** | **0.6280** | **0.6829** |

edits with better consistency. This further demonstrates the impressive performance of our approach in multimodal instruction-based editing.

**Evaluation on Multimodal Instruction-based Image Generation.** As shown in Tab. 3, our method outperforms the commercial model Nano Banana in both human evaluations and assessments by

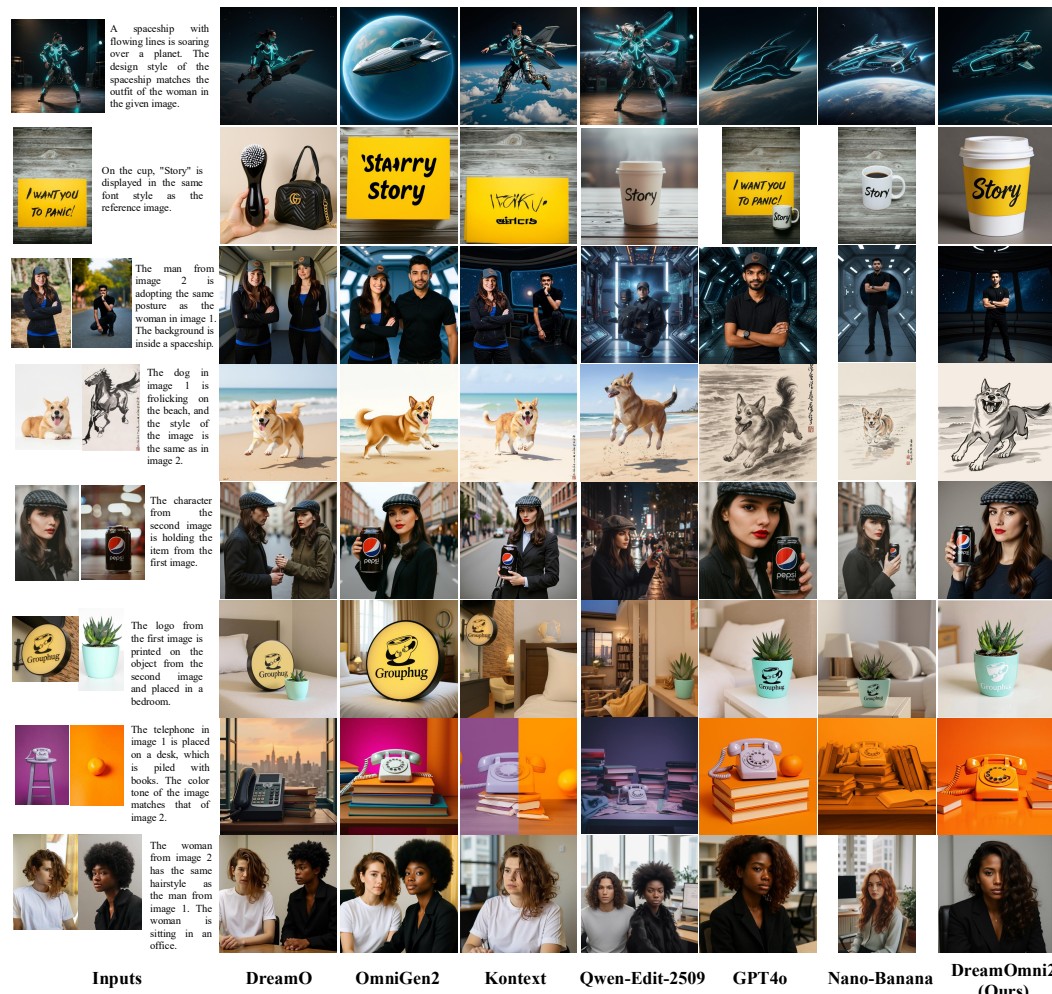

Figure 5: Visual comparison of multimodal instruction-based generation. Our DreamOmni2 significantly outperforms current open-source models and achieves generation results comparable to closed-source commercial models (GPT-4 and Nano Banana).

Doubao 1.6 and Gemini 2.5, achieving results comparable to GPT-4o. Compared to open-source models like DreamO, Omnigen2, and Qwen-Edit-2509, which focus primarily on generating images with multiple concrete objects, DreamOmni2 still significantly outperforms them in both generation accuracy and object consistency, even within their specialized domains. This further underscores the effectiveness of DreamOmni2 in multimodal instruction-based generation.

Quantitative results, as shown in Fig. 5, indicate that open-source models struggle with generating abstract attributes. Even in generating concrete objects, which these models are specifically optimized for, DreamOmni2 outperforms them in both instruction adherence and object consistency. Furthermore, DreamOmni2 even outperforms the commercial model Nano Banana.

**Joint Training.** As shown in Tab. 4, we validate the impact of joint training of generation or editing and VLM. Scheme 1 represents the base model, Kontext. In Scheme 2, we train the generation and editing models with basic instructions without introducing VLM. In Scheme 3, we train the VLM with standard descriptive instructions and input the VLM-generated descriptions into Kontext. In Scheme 4, we perform joint training of the VLM and our generation or editing model on our data. Comparing Scheme 2 with Scheme 1, we see that our data significantly enhances the model's ability to handle multimodal instruction-based editing and generation. Comparing Scheme 3 with Scheme 4, we observe that introducing VLM helps the generation and editing models better understand real-world user complex instructions, improving performance. Moreover, our joint training scheme in Scheme 4 outperforms Scheme 2 and Scheme 3, demonstrating its effectiveness.

Table 3: Quantitative comparison of multimodal instruction-based generation. We use Gemini (Google, 2025a) and Doubao (ByteDance, 2025) to evaluate the success editing ratio on concrete objects and abstract attributions, respectively. In addition, "Human" refers to professional engineers assessing the editing success rates of all models.

| Method | Concrete Object | | | Abstract Attribution | | |
|---|---|---|---|---|---|---|
| | Gemini↑ | Doubao↑ | Human↑ | Gemini↑ | Doubao↑ | Human↑ |
| GPT-4o (OpenAI, 2025) | 0.6250 | 0.6250 | 0.5610 | 0.6889 | 0.6333 | 0.5793 |
| Nano Banana (Google, 2025b) | 0.5000 | 0.5417 | 0.5366 | 0.5556 | 0.5111 | 0.3293 |
| UNO (Wu et al., 2025c) | 0.0000 | 0.0000 | 0.0000 | 0.0333 | 0.0556 | 0.0000 |
| DreamO (Mou et al., 2025) | 0.0417 | 0.0833 | 0.0000 | 0.0667 | 0.0222 | 0.0000 |
| Omnigen2 (Wu et al., 2025b) | 0.2083 | 0.2500 | 0.2927 | 0.1000 | 0.0778 | 0.0305 |
| Qwen-Image-Edit (Wu et al., 2025a) | 0.0417 | 0.1250 | 0.0244 | 0.0889 | 0.1000 | 0.0000 |
| Kontext (Batifol et al., 2025) | 0.2500 | 0.3750 | 0.0976 | 0.0556 | 0.1222 | 0.0122 |
| Qwen-Image-Edit-2509 (Wu et al., 2025a) | 0.1250 | 0.2917 | 0.2195 | 0.1111 | 0.1556 | 0.0427 |
| DreamOmni2 (Ours) | **0.5833** | **0.6667** | **0.6098** | **0.5778** | **0.6333** | **0.6829** |

Table 4: The validation of joint training for generation or editing models and VLM.

| Method | Generation or Editing Model Training | VLM Training | Editing | | Generation | |
|---|---|---|---|---|---|---|
| | | | Concrete Object | Abstract Attribution | Concrete Object | Abstract Attribution |
| Scheme 1 | ✗ | ✗ | 0.1220 | 0.0122 | 0.3750 | 0.1222 |
| Scheme 2 | ✓ | ✗ | 0.3659 | 0.3171 | 0.4583 | 0.3444 |
| Scheme 3 | ✗ | ✓ | 0.2439 | 0.3415 | 0.5417 | 0.4778 |
| Scehme 4 (Ours) | ✓ | ✓ | 0.6585 | 0.6280 | 0.6667 | 0.6333 |

**Index and Position Encoding.** As shown in Tab. 5, we compare different encoding schemes to help the model adapt to multiple image inputs. Comparing Scheme 3 and Scheme 1, we find that adding index encoding enables the model to understand which image corresponds to references like "Image 1," "Image 2," and "Image 3" in user instructions, resulting in more accurate generation and editing. Additionally, when comparing Scheme 3 and Scheme 4, we observe that with the inclusion of index encoding, multiple images require position encoding shifts instead of using the same position encoding. This adjustment prevents the copy-and-paste effect and improves the model's editing and generation performance. Therefore, in DreamOmni2, we incorporate index encoding along with position encoding shifts for multiple reference images.

Table 5: The validation of different encoding schemes for multiple image inputs.

| Method | Index Encoding | Position Encoding Shift | Editing | | Generation | |
|---|---|---|---|---|---|---|
| | | | Concrete Object | Abstract Attribution | Concrete Object | Abstract Attribution |
| Scehme 1 | ✗ | ✗ | 0.2439 | 0.2805 | 0.2917 | 0.2222 |
| Scehme 2 | ✗ | ✓ | 0.4634 | 0.5427 | 0.5417 | 0.5111 |
| Scehme 3 | ✓ | ✗ | 0.3415 | 0.3902 | 0.4167 | 0.4556 |
| Scheme 4 (Ours) | ✓ | ✓ | 0.6585 | 0.6280 | 0.6667 | 0.6333 |

## 5 CONCLUSION

Current instruction-based editing relies on language, but it often struggles to clearly describe desired edits. Therefore, reference images are needed to guide the process. Additionally, subject-driven generation models typically focus on concrete objects and cannot generate images based on abstract concepts. To this end, we propose two new tasks: multimodal instruction-based editing and generation, where references include both concrete objects and abstract attributions. These tasks face two main challenges: training data and the framework supporting multi-image input. For training data, we introduce a three-stage data synthesis pipeline. In stage 1, we use a feature mixing approach to create data for an extraction model, which can generate images with the same elements (objects or attributes) as the given image. In stage 2, we use the extraction and instruction-based editing models to create multimodal instruction-based editing data. In stage 3, we apply the extraction model to stage 2 data to generate multimodal instruction-based generation data. For the framework, we design an index encoding and position encoding shift scheme to help the model distinguish multiple images and avoid the copy-and-paste effect. We also propose a joint training scheme for the generation/editing model and the VLM, improving the model's ability to understand complex real-world instructions. Extensive experiments show the impressive performance of DreamOmni2.

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

# A APPENDIX

## A.1 LLM-USAGE STATEMENT

The authors used a large language model to assist with language polishing, grammar correction, and typo identification in this paper. The ideas, methodology, experimental design, and results presented are the sole work of the authors.

## A.2 ETHICS STATEMENT

This research follows the ICLR Code of Ethics. The primary contribution of our work is the proposal of two new tasks, multimodal instruction-based editing and generation, which enable users to create more easily. We acknowledge the importance of the responsible application of this technology. Our research does not involve the collection or use of any new personally identifiable information, and all experiments were conducted on publicly available and synthetic datasets.

## A.3 REPRODUCIBILITY STATEMENT

Our work is designed to be fully reproducible. We have provided detailed descriptions of our methodology, including the model architectures, training procedures, and hyperparameter settings, within the main text and supplementary materials of this paper. Comprehensive ablation studies are also included to demonstrate the robustness and influence of key components of our proposed approach. The code and models in this paper have been open-sourced.

## A.4 DREAMOMNI2 BENCHMARK

Our DreamOmni2 benchmark includes 205 multimodal instruction-based editing test cases and 114 instruction-based generation test cases. Visualizations of the editing and generation test cases are shown in Fig. 7 and Fig. 6, respectively. The benchmark covers a wide range of test cases, with input reference images ranging from one to five, and encompasses diverse local and global attributes, as well as concrete objects. The DreamOmni2 Benchmark will be released.

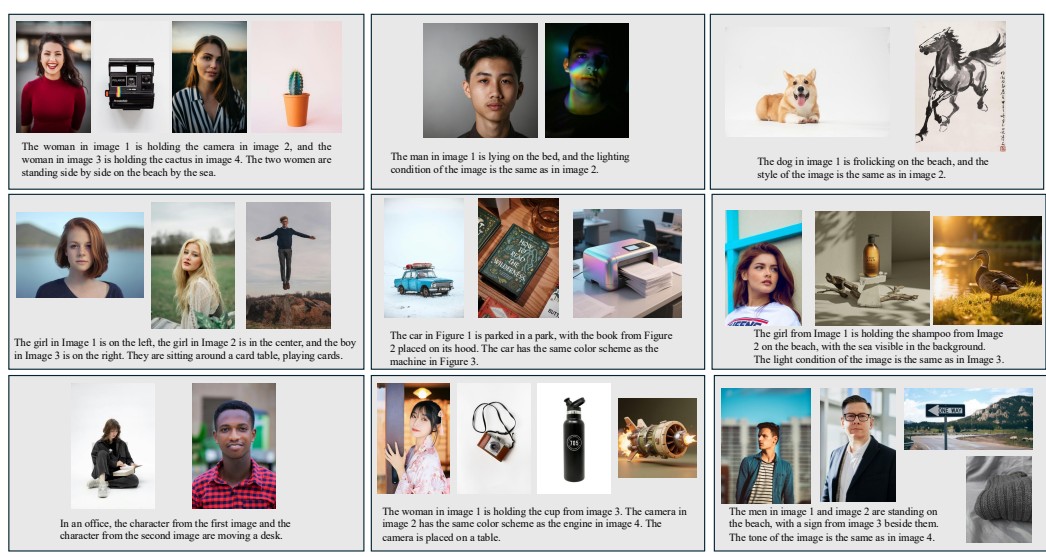

Figure 6: Examples of multimodal instruction-based generation in DreamOmni2 benchmark.

## A.5 MORE MULTIMODAL INSTRUCTION-BASED EDITING CASES

As shown in Fig. 8, Fig. 9, Fig. 10, Fig. 11, Fig. 12, Fig. 13, Fig. 14, Fig. 15, Fig. 16, Fig. 17, Fig. 18, Fig. 19, Fig. 20, and Fig. 21, we present additional visual cases of DreamOmni2 on the multimodal instruction-based editing task.

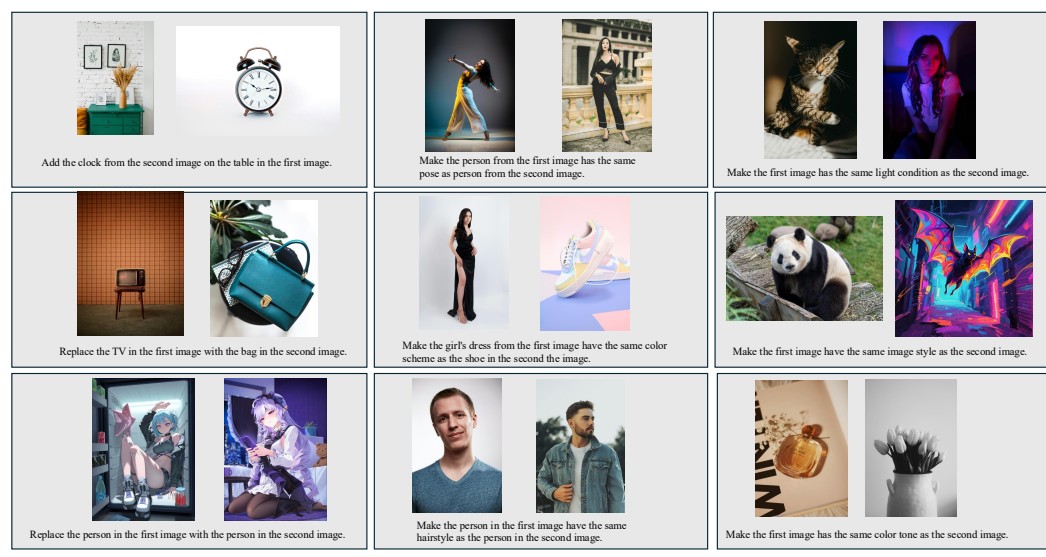

| Concrete Objects | Local Attribution | Global Attribution |

Figure 7: Examples of multimodal instruction-based editing in DreamOmni2 benchmark.

## A.6   MORE MULTIMODAL INSTRUCTION-BASED GENERATION CASES

As shown in Fig. 22, Fig. 23, Fig. 24, Fig. 25, Fig. 26, Fig. 27, Fig. 28, Fig. 29, and Fig. 30, we present additional visual cases of DreamOmni2 on the multimodal instruction-based generation task.

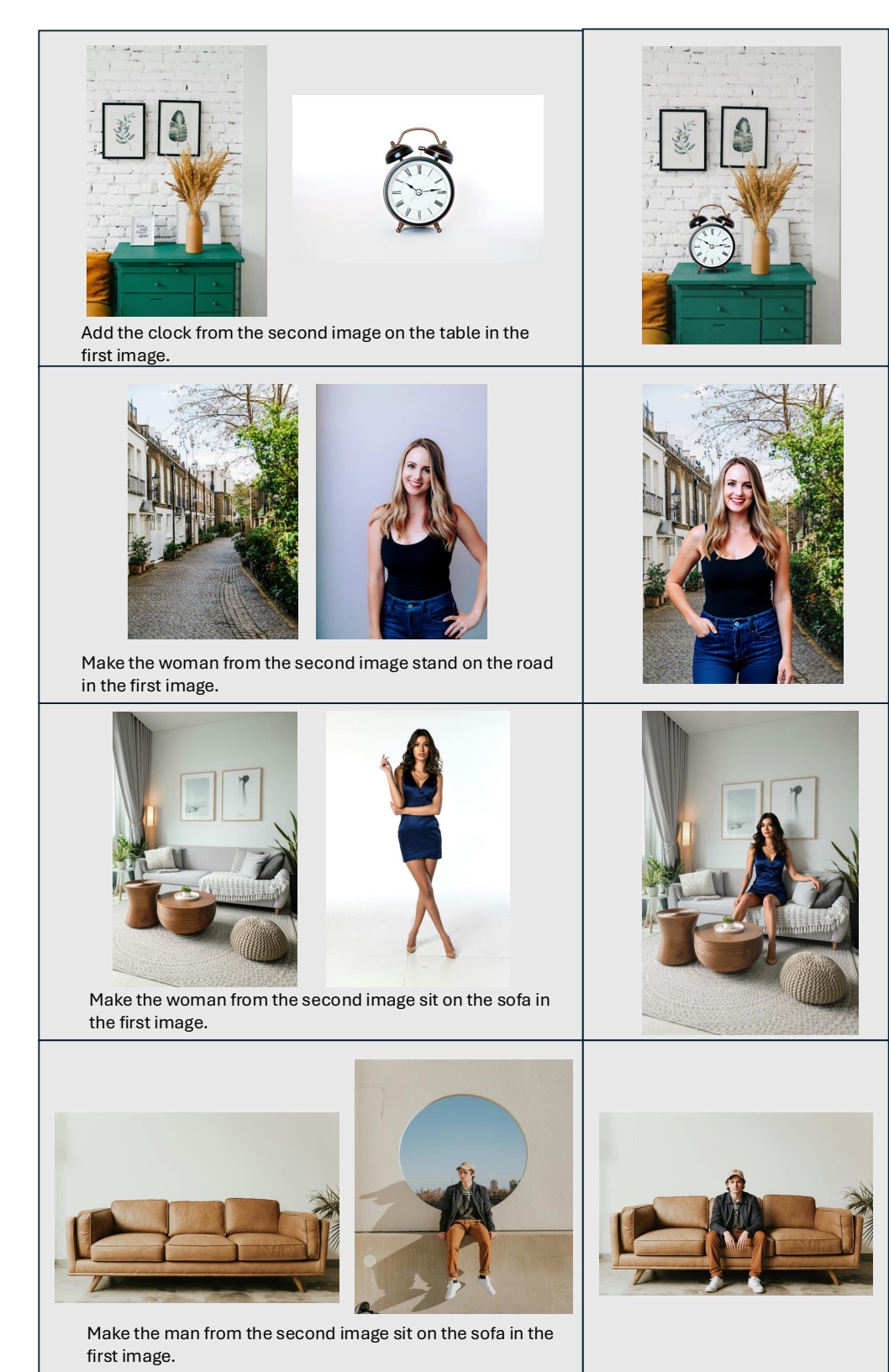

Figure 8: Multimodal instruction-based editing cases of DreamOmni2.

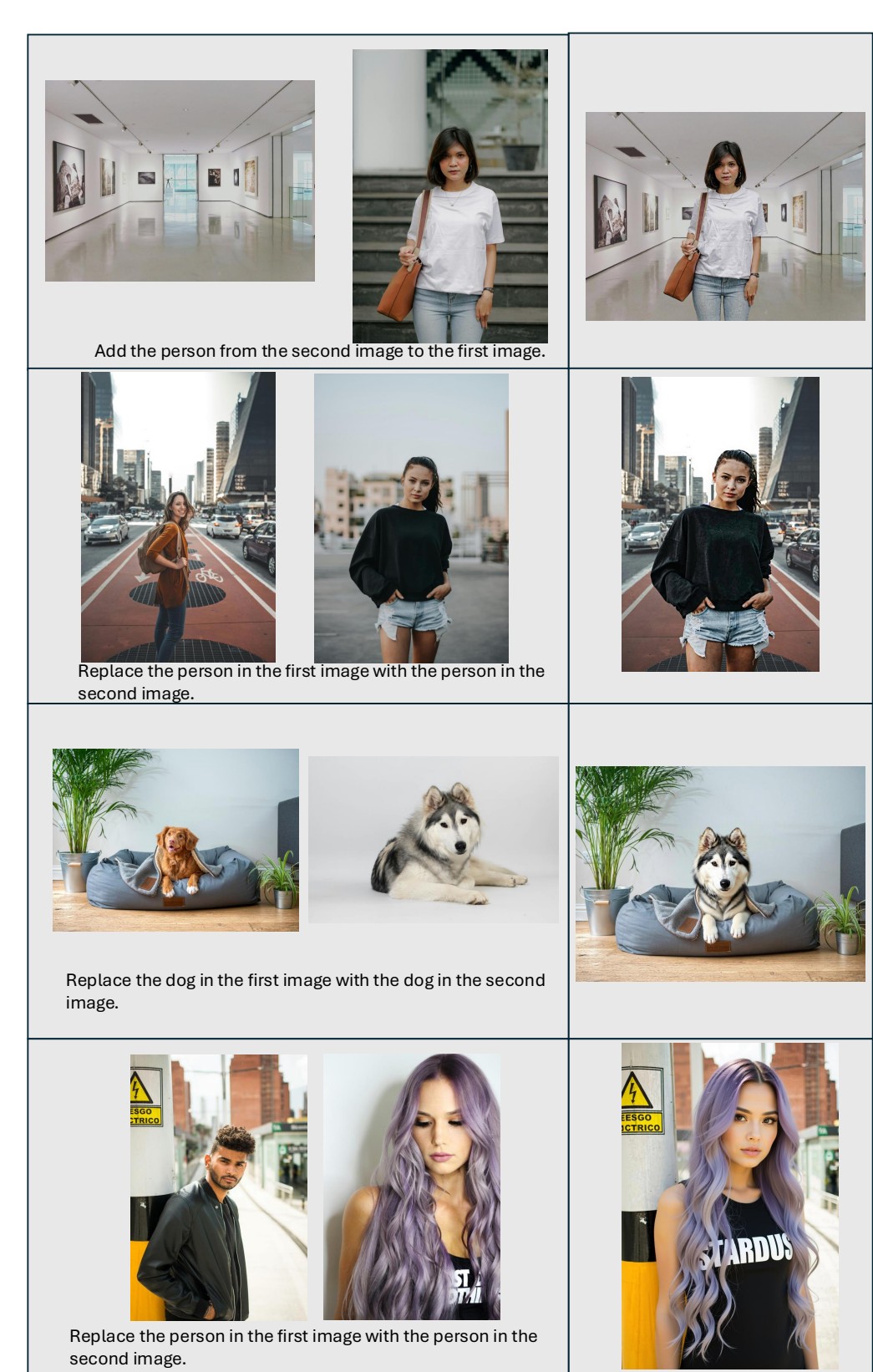

Figure 9: Multimodal instruction-based editing cases of DreamOmni2.

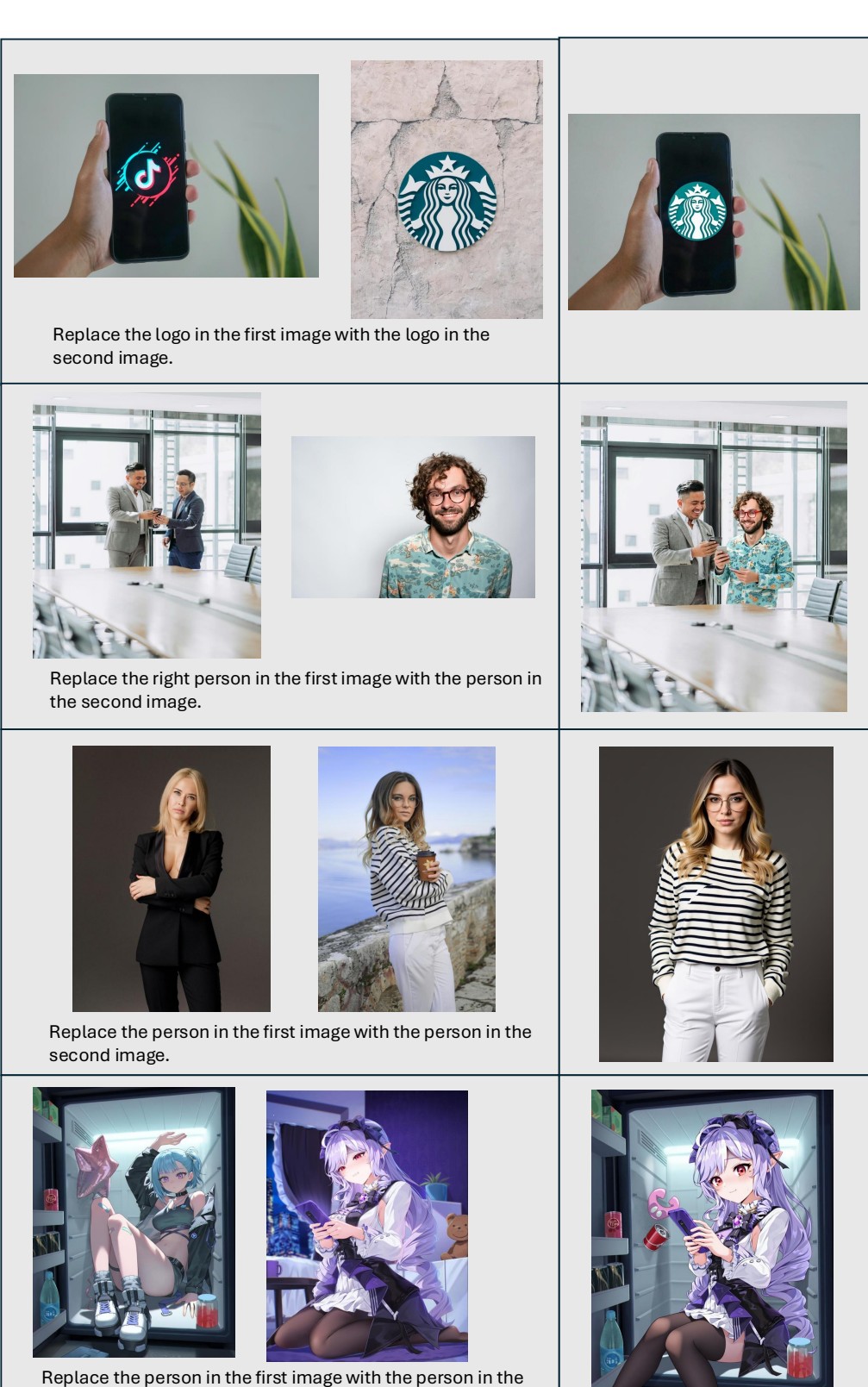

Figure 10: Multimodal instruction-based editing cases of DreamOmni2.

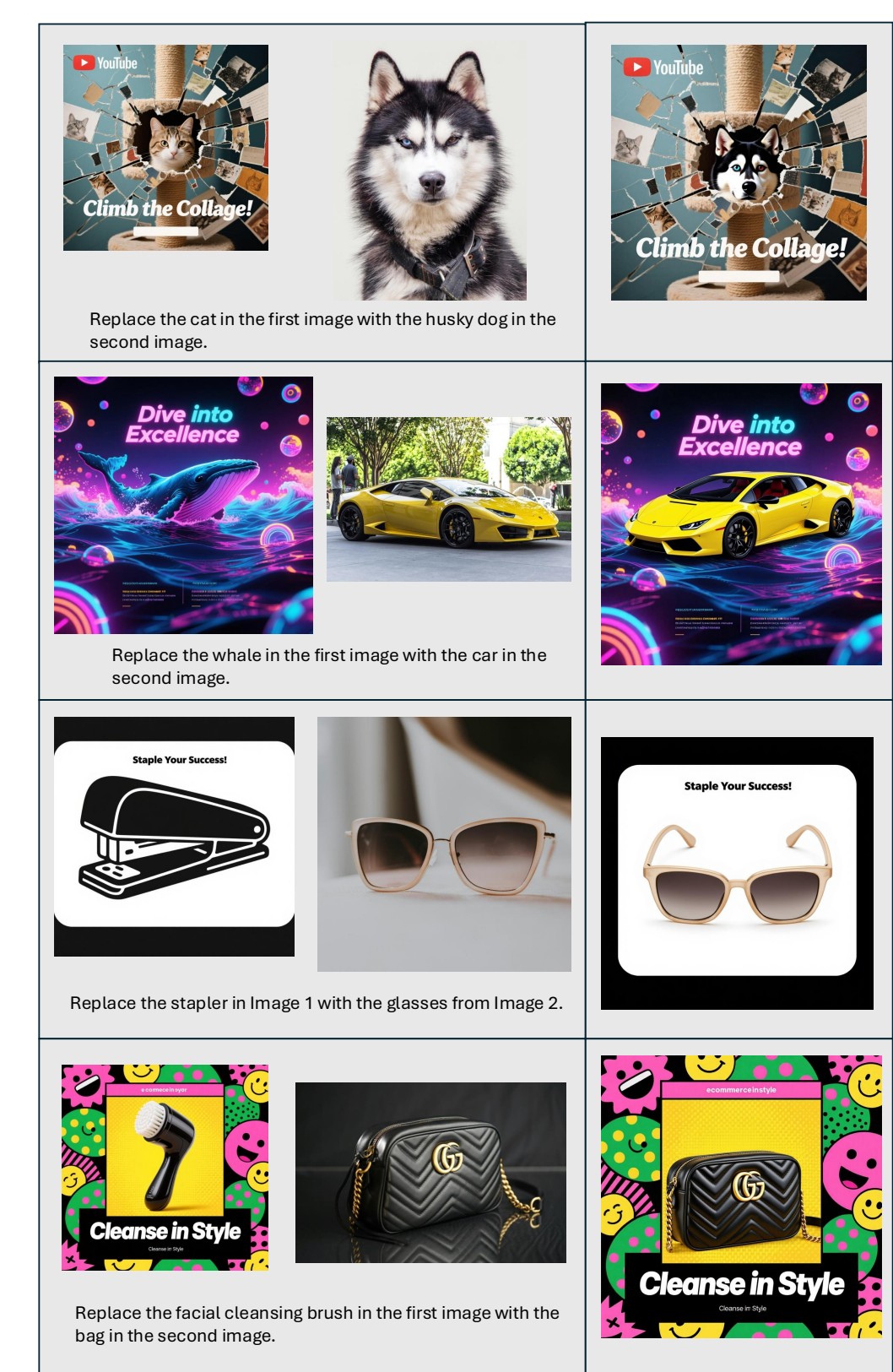

Figure 11: Multimodal instruction-based editing cases of DreamOmni2.

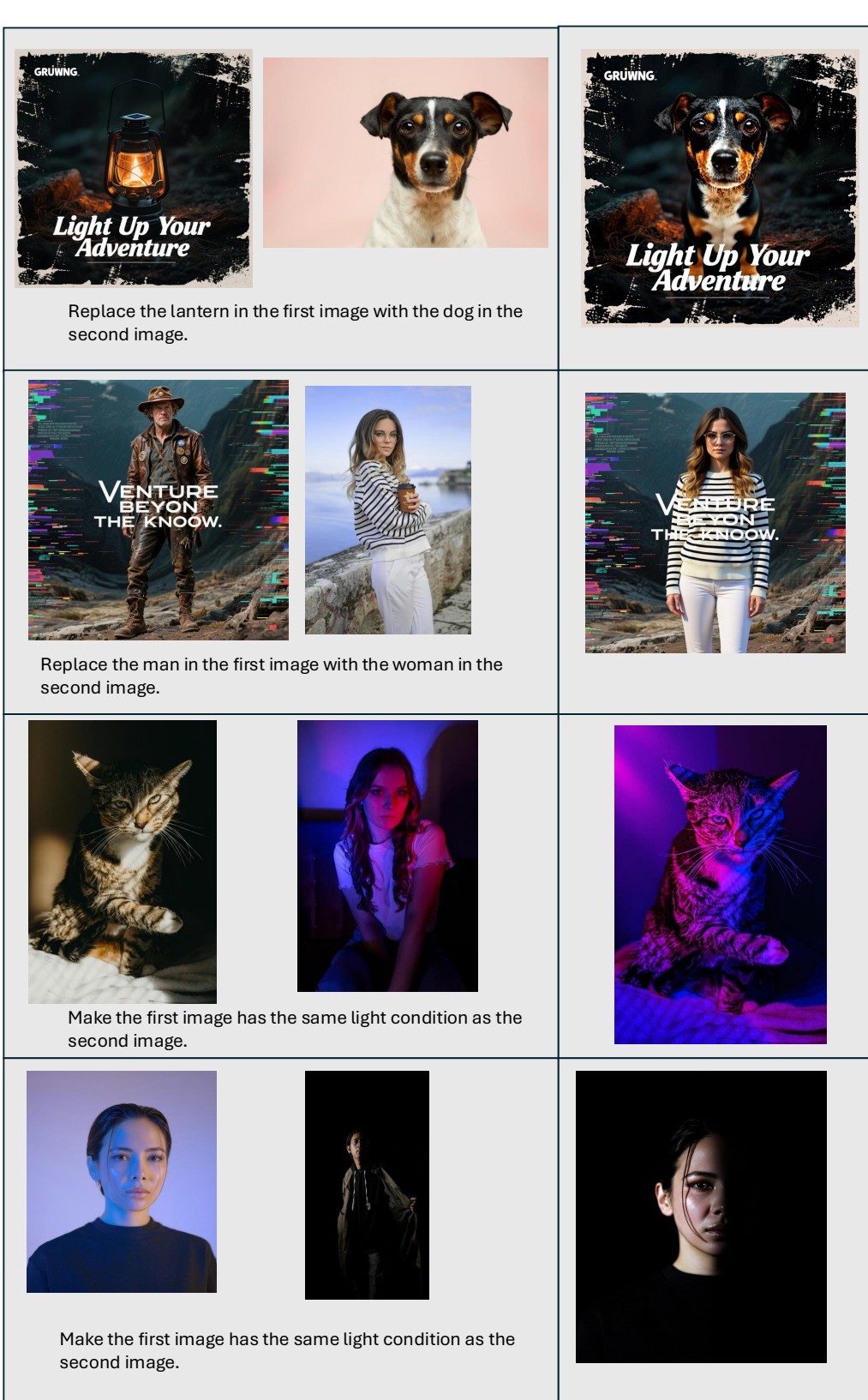

Figure 12: Multimodal instruction-based editing cases of DreamOmni2.

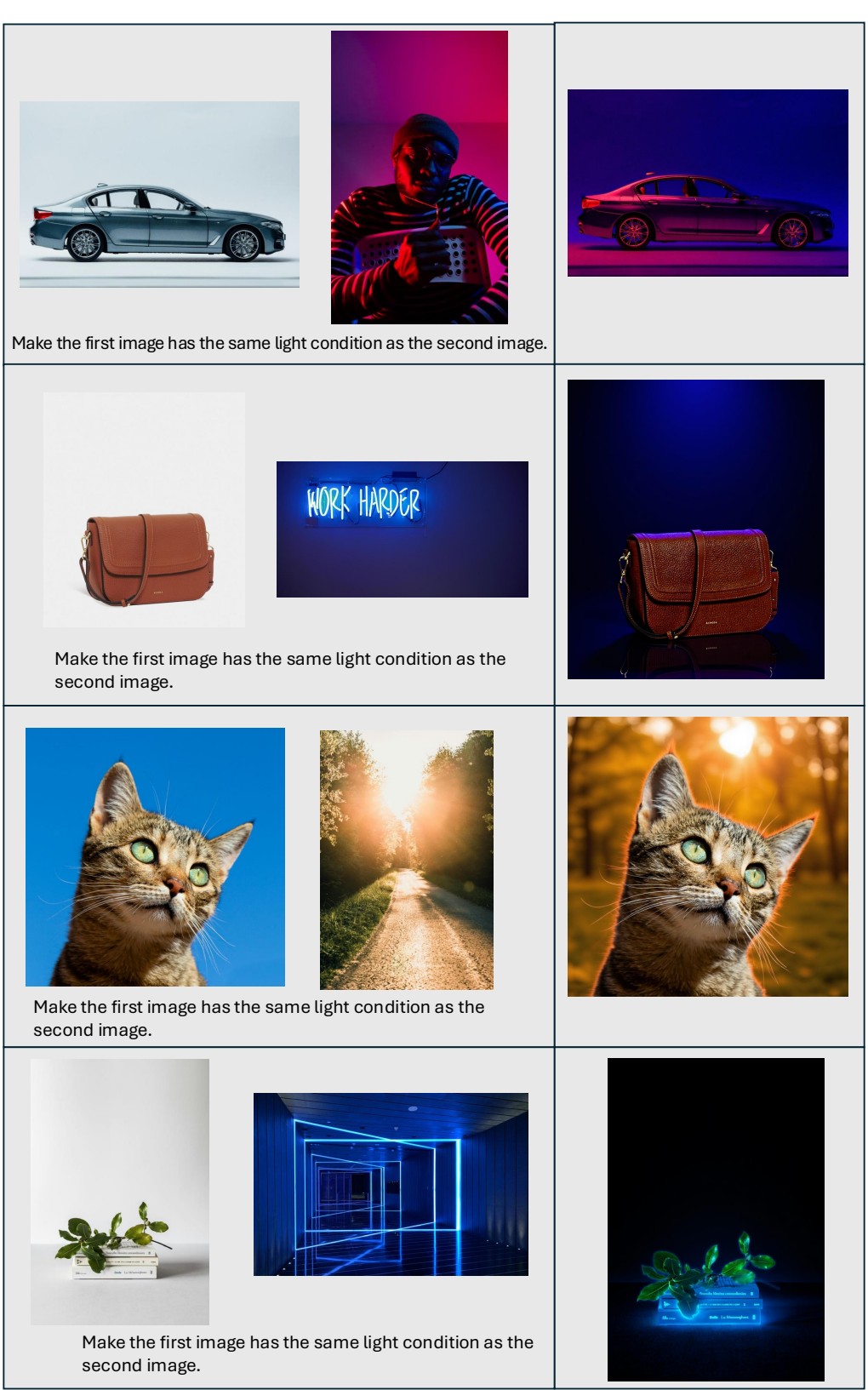

Figure 13: Multimodal instruction-based editing cases of DreamOmni2.

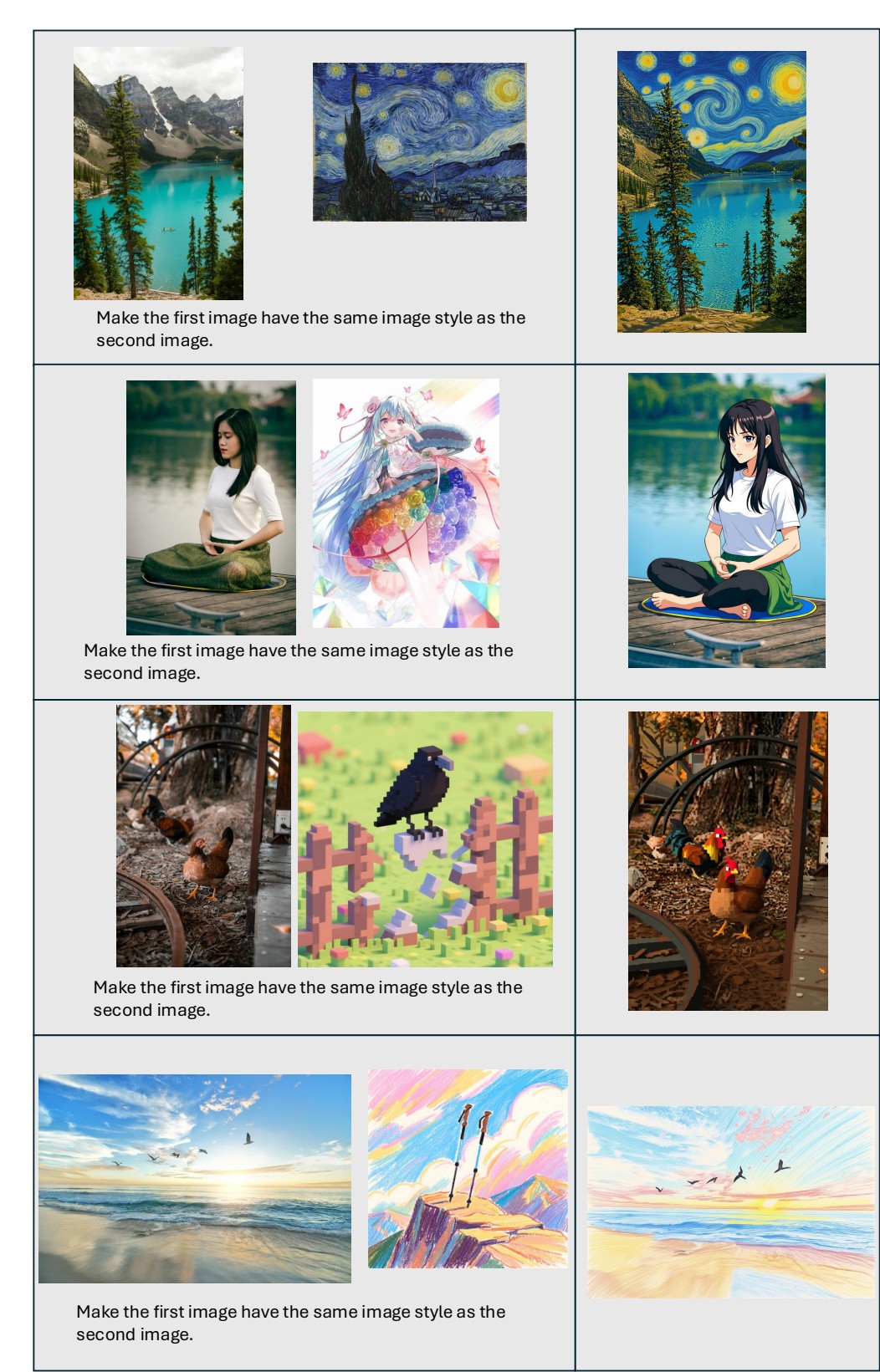

Figure 14: Multimodal instruction-based editing cases of DreamOmni2.

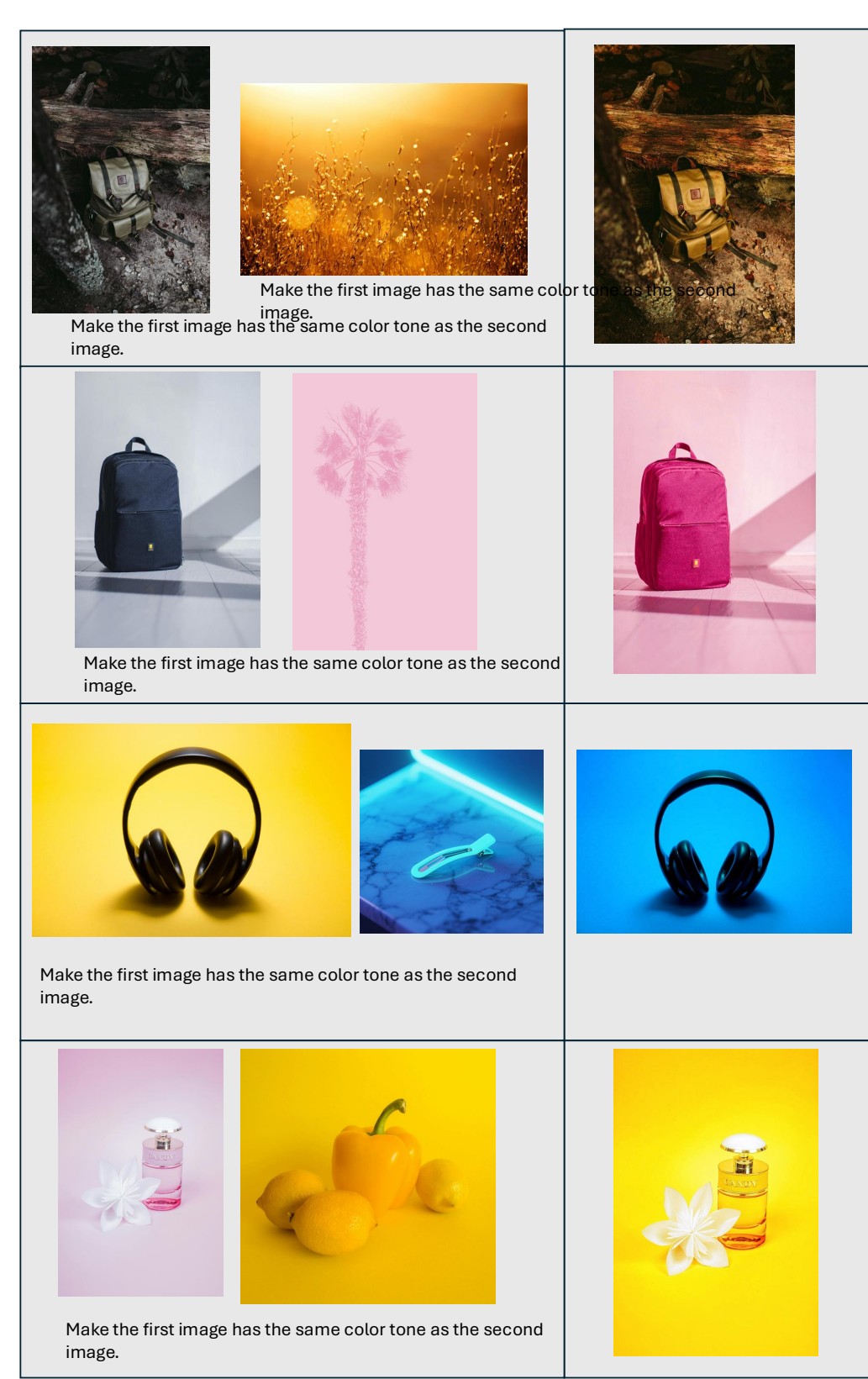

Figure 15: Multimodal instruction-based editing cases of DreamOmni2.

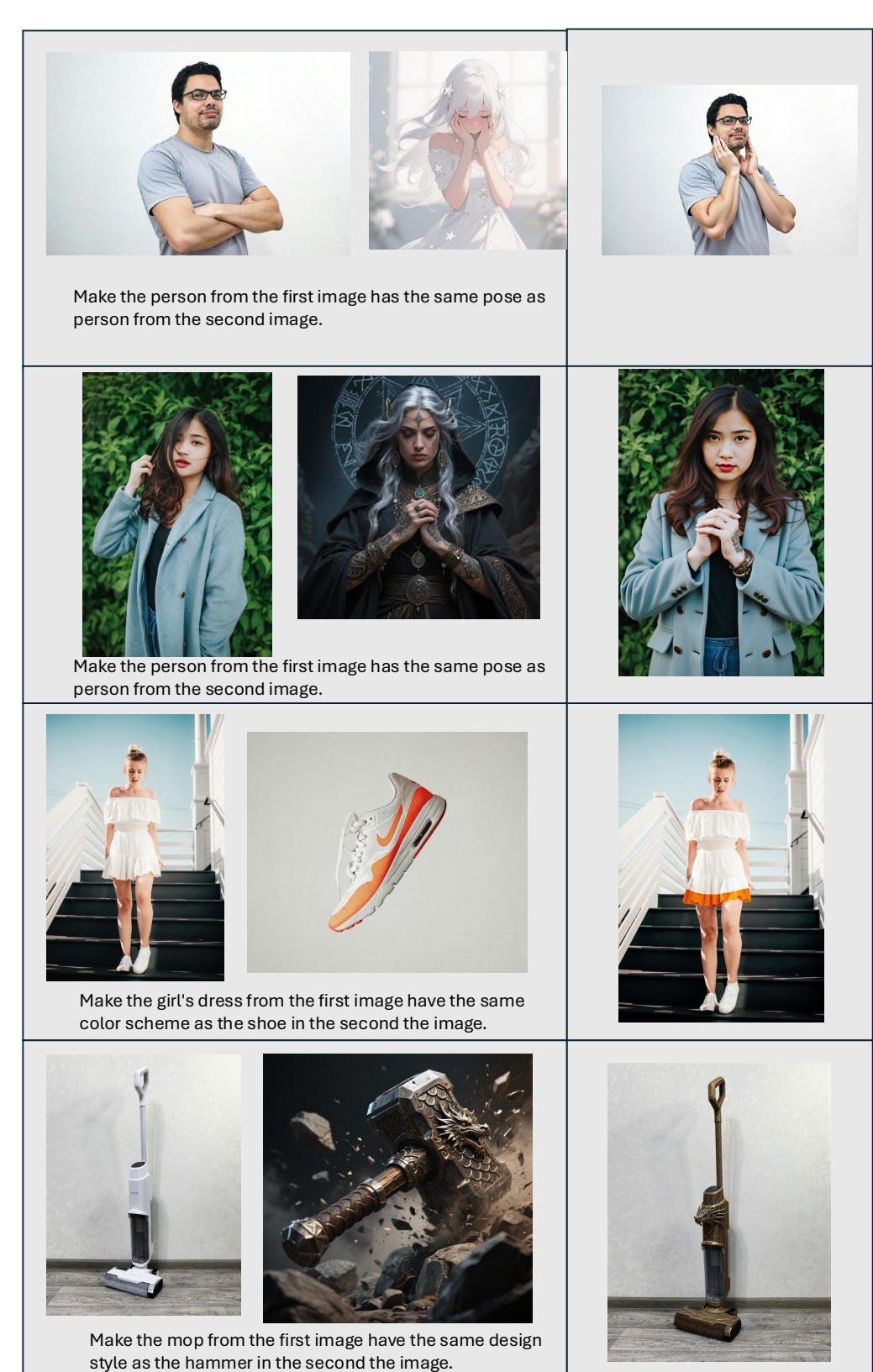

Figure 16: Multimodal instruction-based editing cases of DreamOmni2.

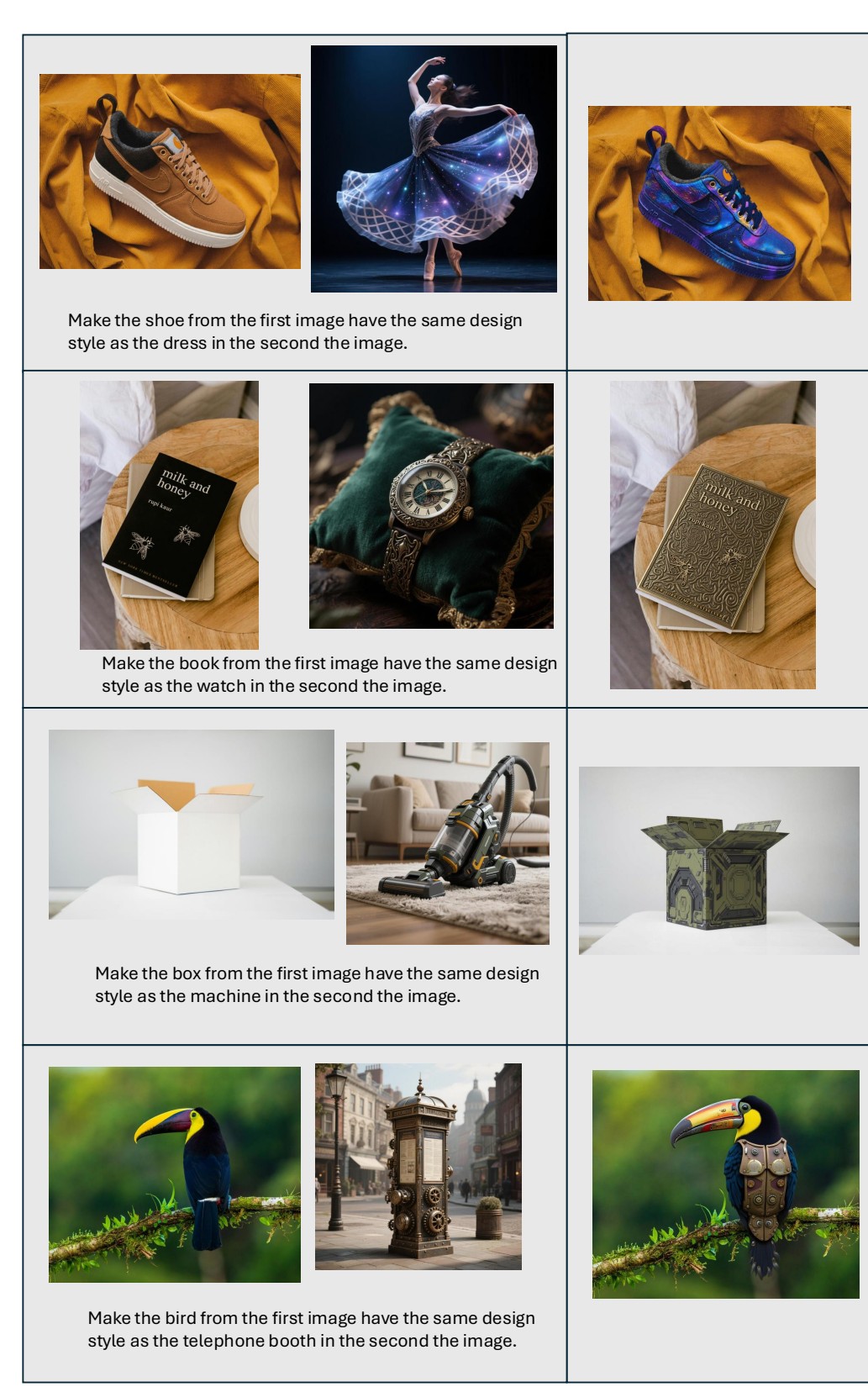

Figure 17: Multimodal instruction-based editing cases of DreamOmni2.

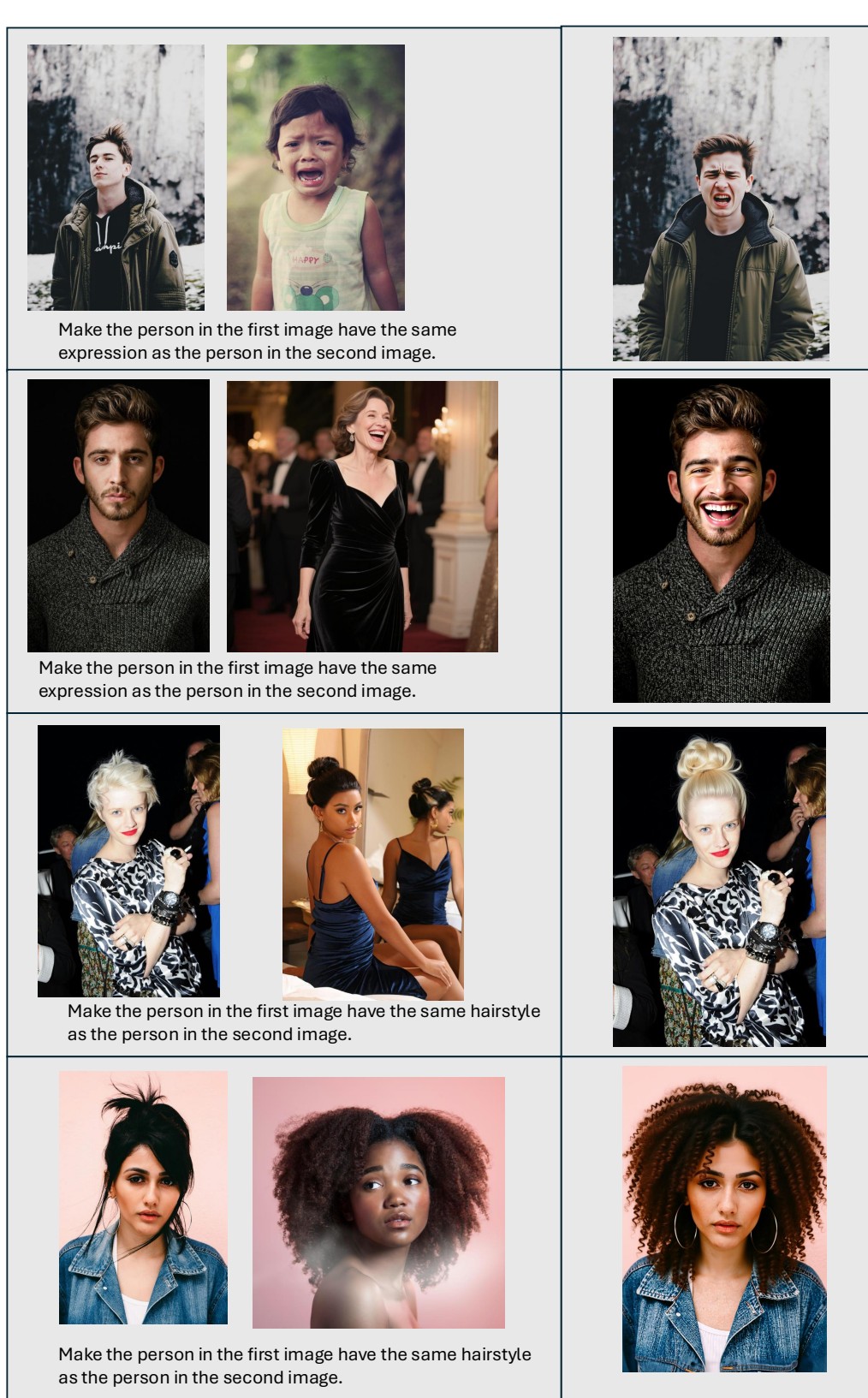

Figure 18: Multimodal instruction-based editing cases of DreamOmni2.

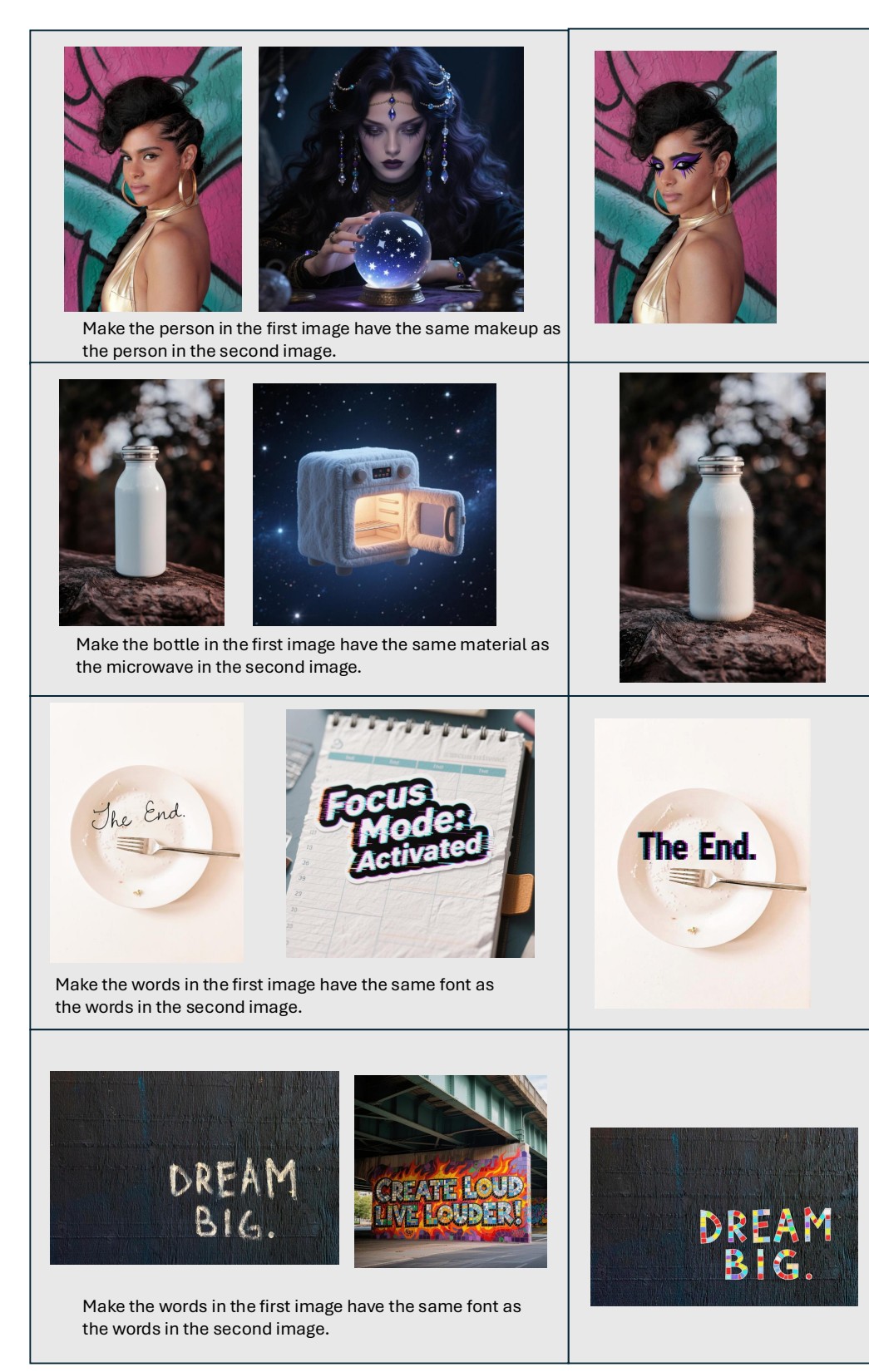

Figure 19: Multimodal instruction-based editing cases of DreamOmni2.

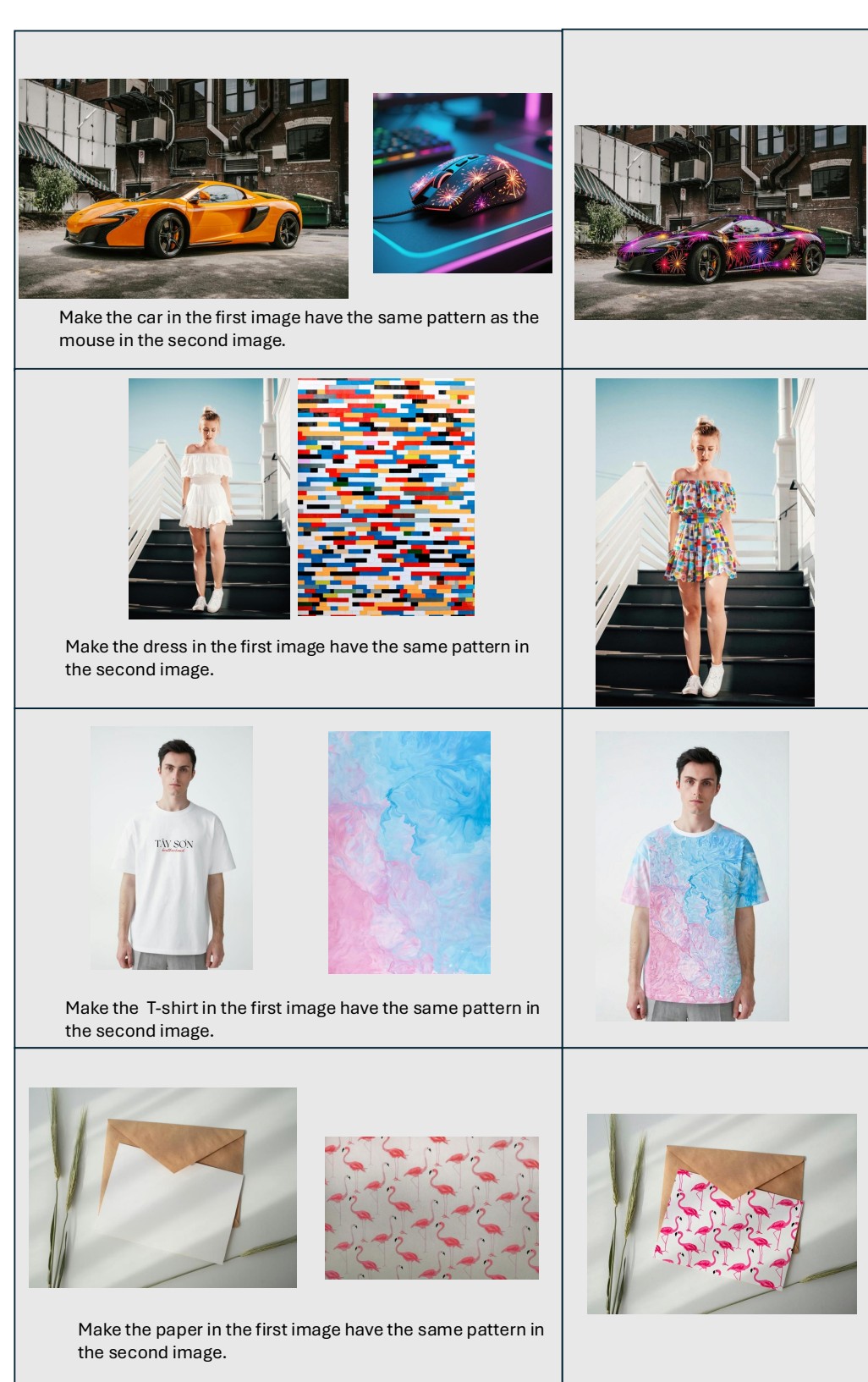

Figure 20: Multimodal instruction-based editing cases of DreamOmni2.

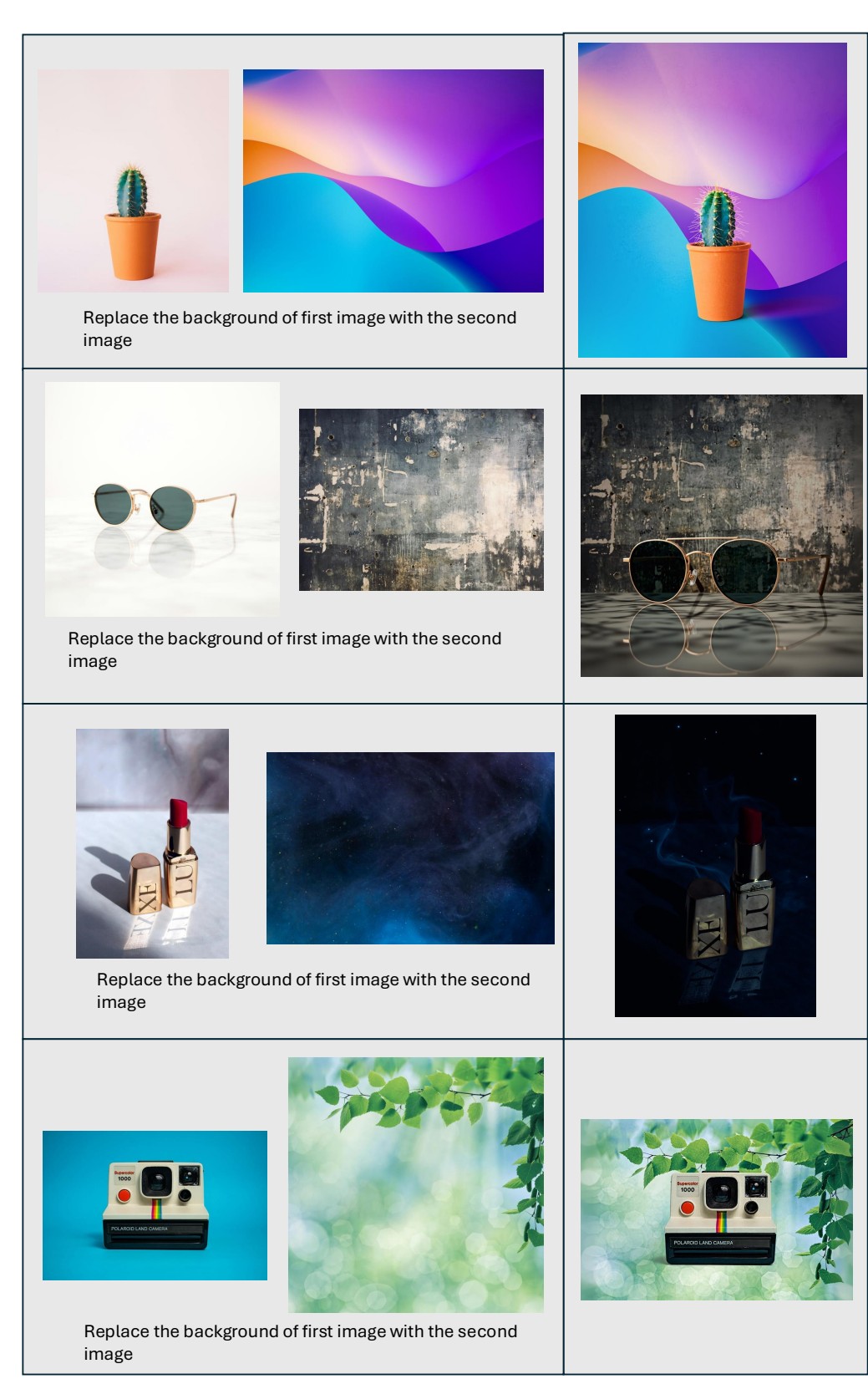

Figure 21: Multimodal instruction-based editing cases of DreamOmni2.

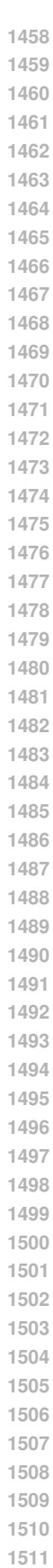

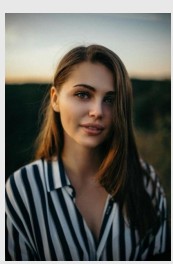 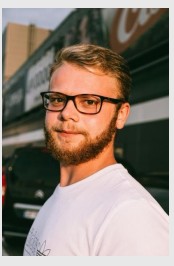

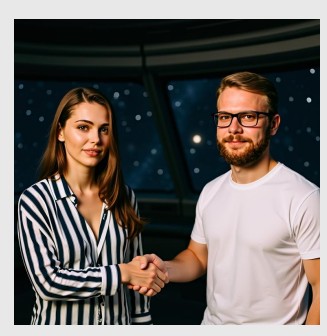

In the scene, the character from the first image stands on the left, and the character from the second image stands on the right. They are shaking hands against the backdrop of a spaceship interior.

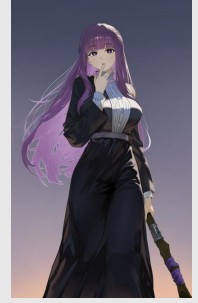 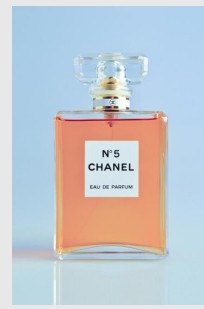

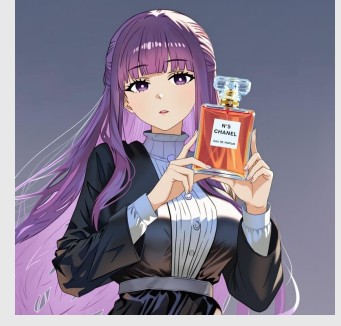

The character from the first image is holding the item from the second picture.

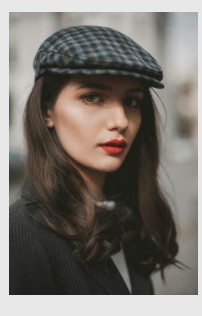 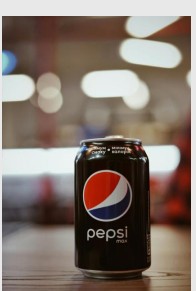

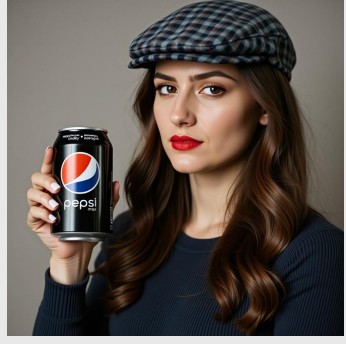

The character from the second image is holding the item from the first image.

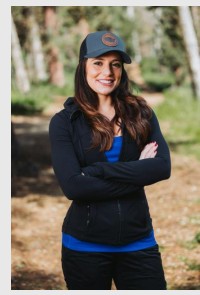 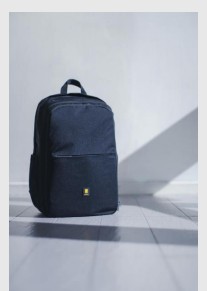

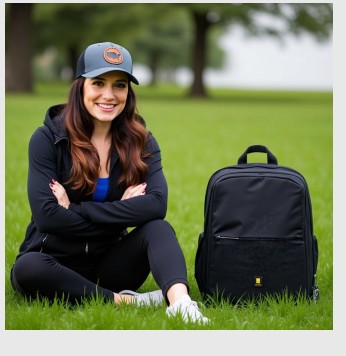

On a lawn, the woman from the first image is sitting on the grass, and the backpack from the second image is placed beside her.

Figure 22: Multimodal instruction-based generation cases of DreamOmni2.

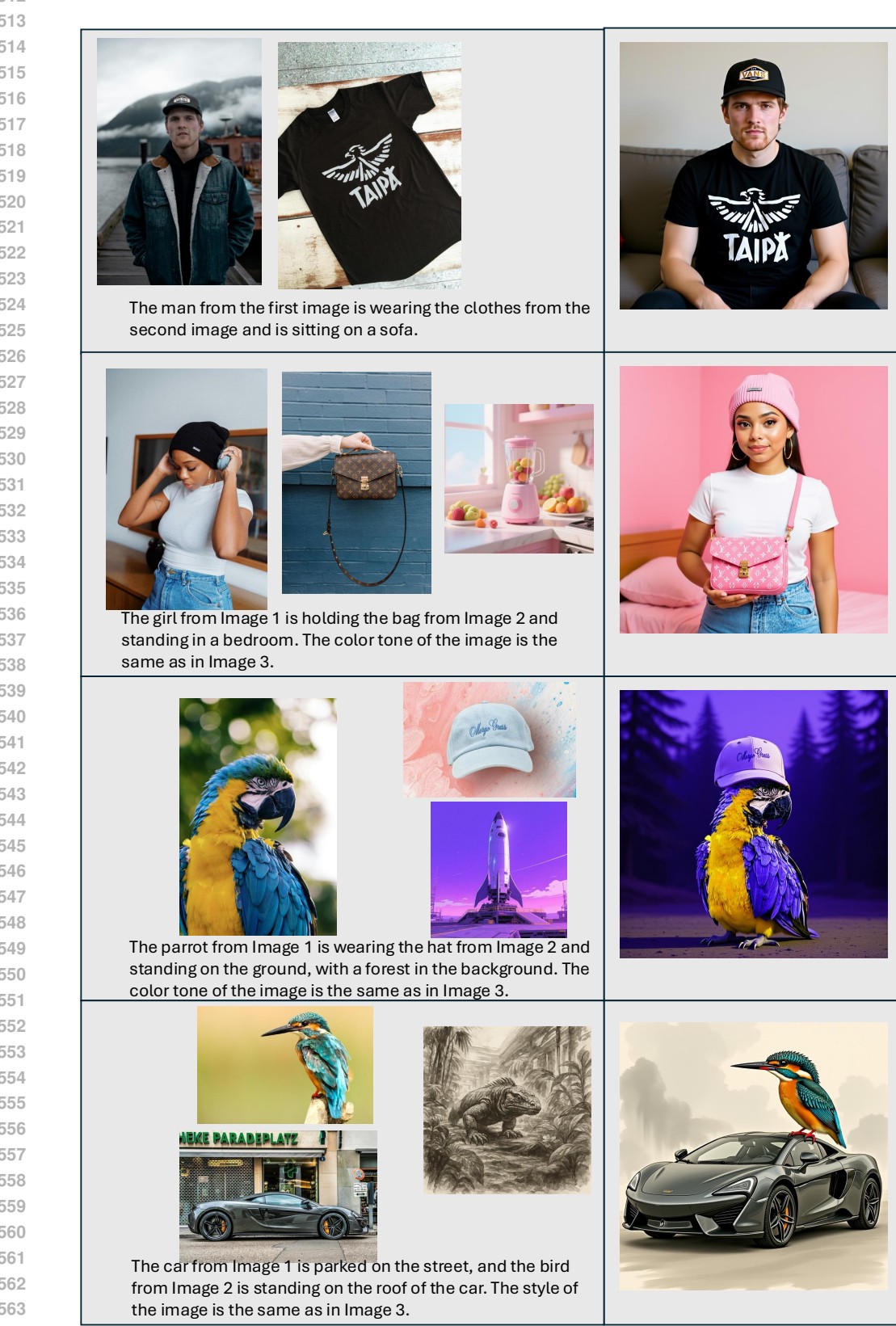

Figure 23: Multimodal instruction-based generation cases of DreamOmni2.

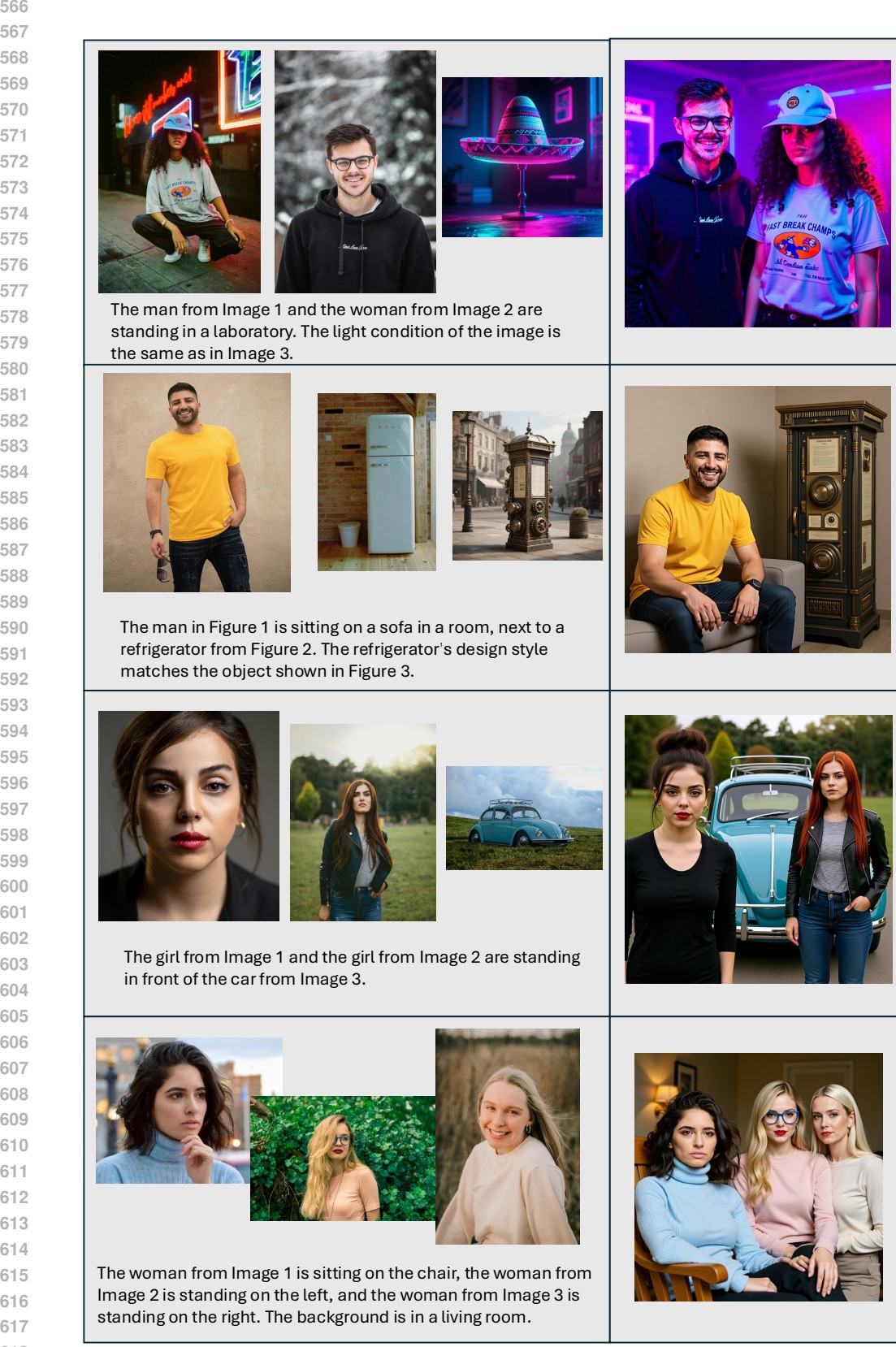

The man from Image 1 and the woman from Image 2 are standing in a laboratory. The light condition of the image is the same as in Image 3.

The man in Figure 1 is sitting on a sofa in a room, next to a refrigerator from Figure 2. The refrigerator's design style matches the object shown in Figure 3.

The girl from Image 1 and the girl from Image 2 are standing in front of the car from Image 3.

The woman from Image 1 is sitting on the chair, the woman from Image 2 is standing on the left, and the woman from Image 3 is standing on the right. The background is in a living room.

Figure 24: Multimodal instruction-based generation cases of DreamOmni2.

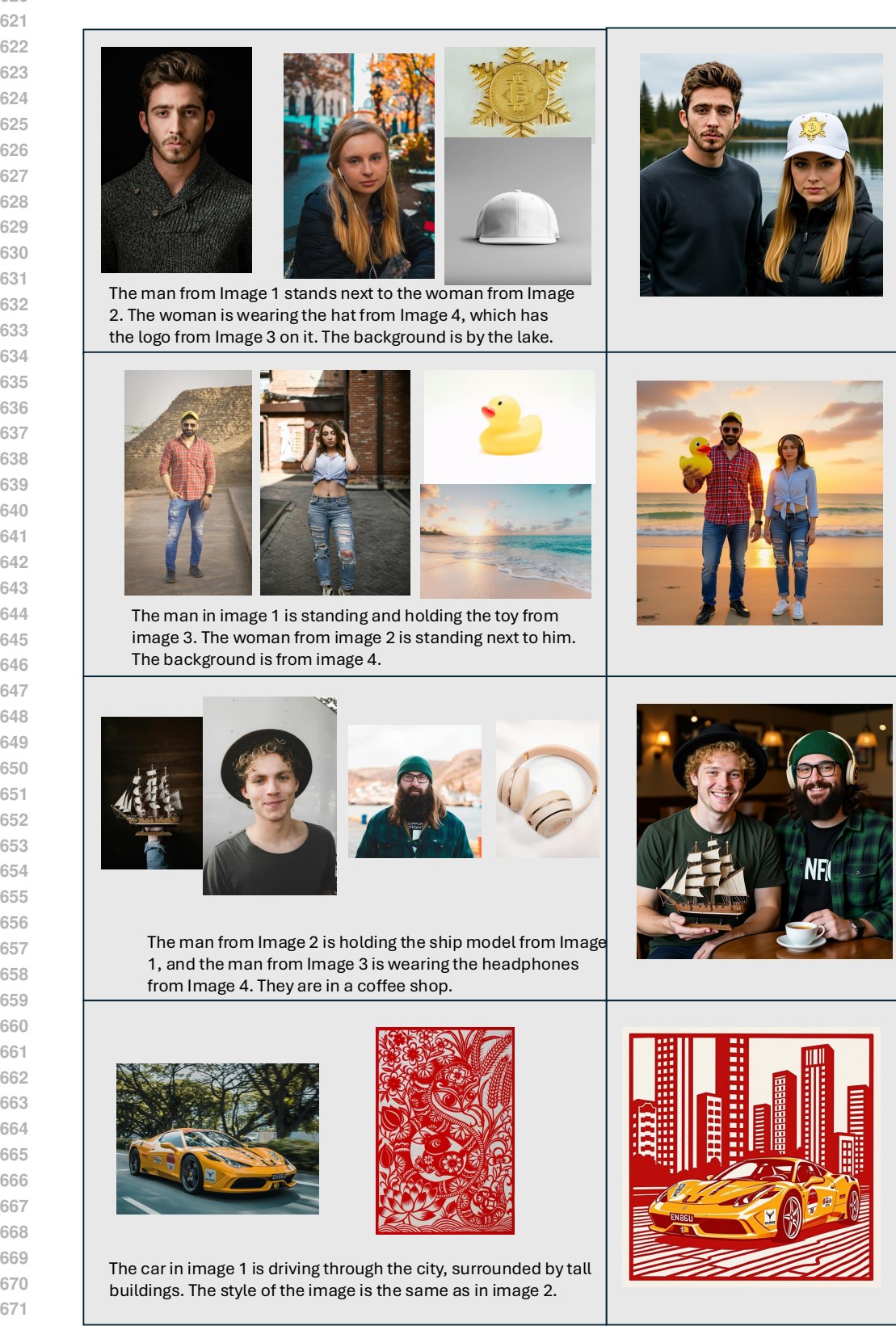

Figure 25: Multimodal instruction-based generation cases of DreamOmni2.

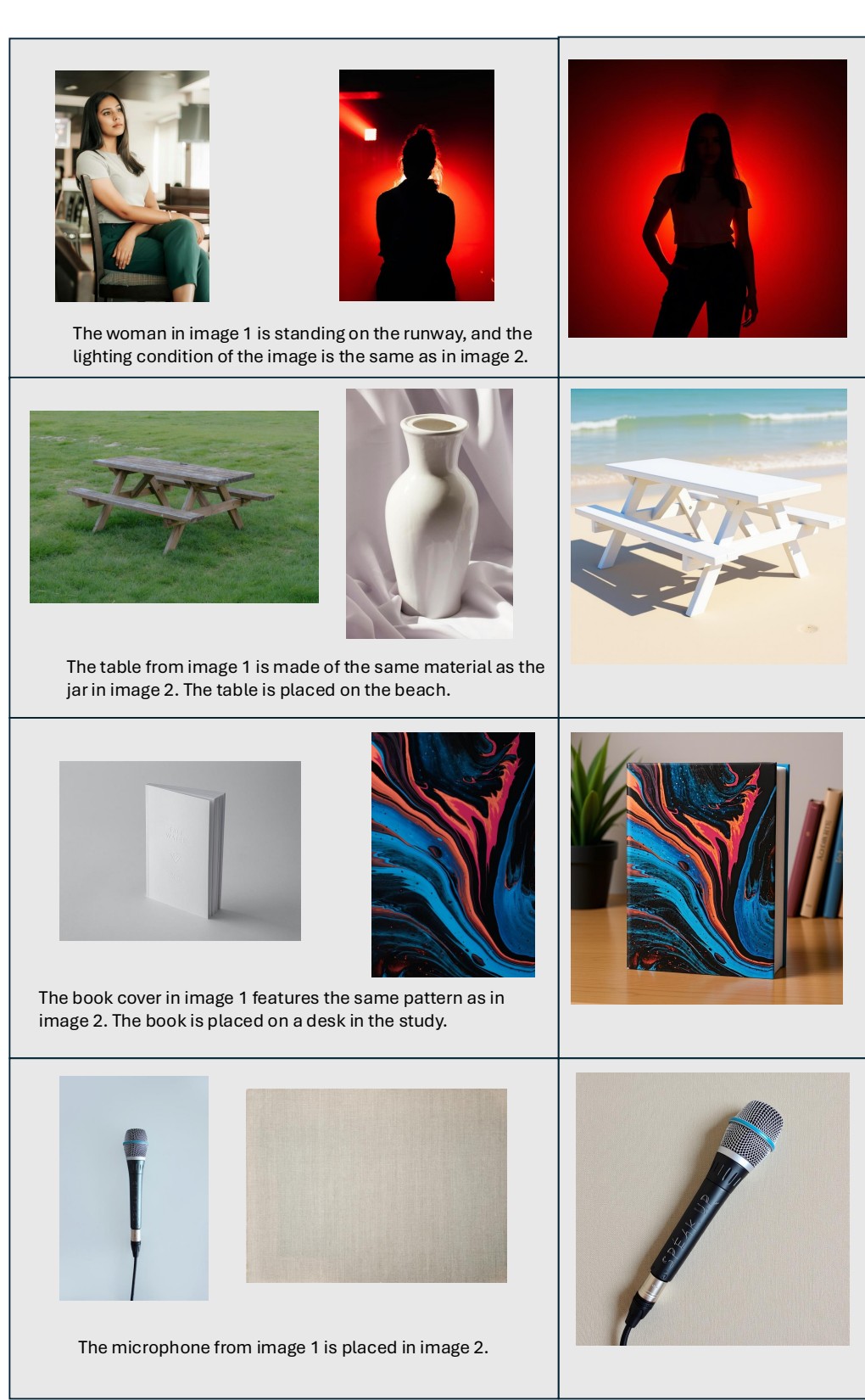

Figure 26: Multimodal instruction-based generation cases of DreamOmni2.

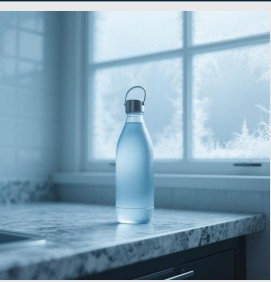
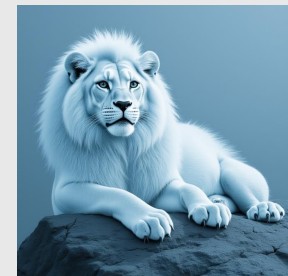

A majestic lion resting on a rocky outcrop. The color tone of the image is the same as in Image 1.

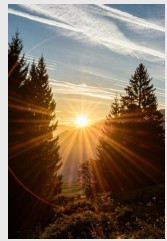
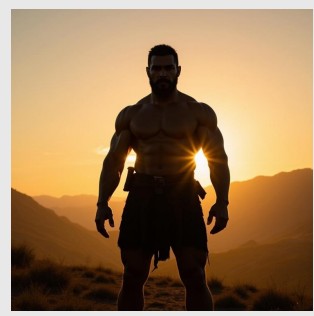

A warrior stands on the battlefield. The lighting conditions of the image are the same as in the reference image.

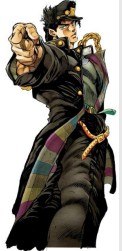
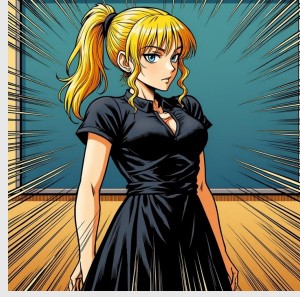

A blonde girl with a high ponytail, wearing a black long dress, stands at the front of the classroom. The style of the image is the same as the given image.

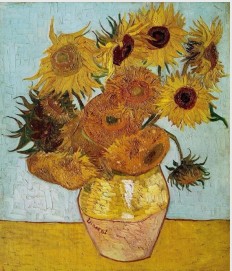
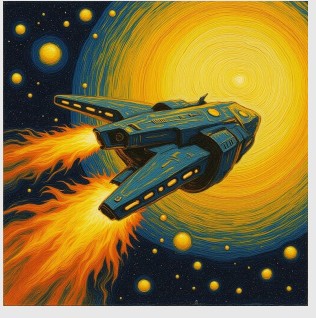

A spaceship is flying in the sky, with the sun visible in the background. The style of the image is the same as in Image 1.

Figure 27: Multimodal instruction-based generation cases of DreamOmni2.

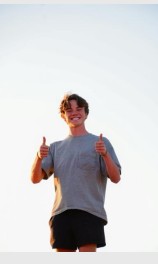

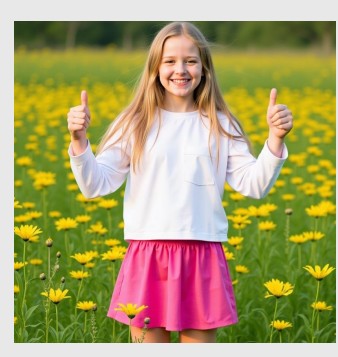

A girl wearing a pink skirt and a white long-sleeve shirt, with long golden hair. She strikes the same pose as the man in the given image. The background is a field of flowers.

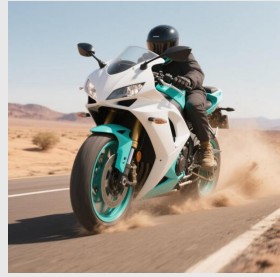

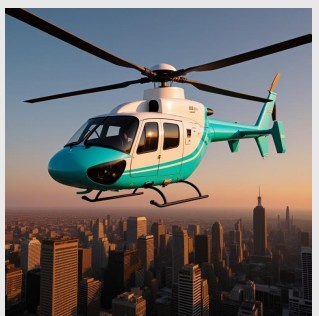

Generate a helicopter soaring above a city skyline at dusk. The color scheme of the helicopter is the same as that of the motorcycle.

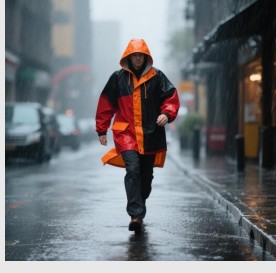

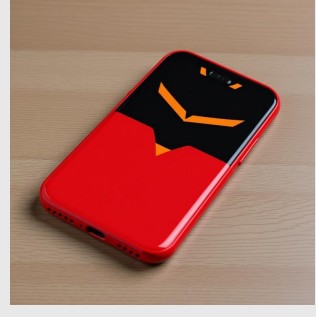

A sleek smartphone resting on a table, with its design featuring smooth curves and a modern look. The color scheme of the phone matches the outfit of the man in the reference image.

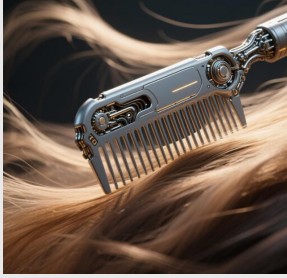

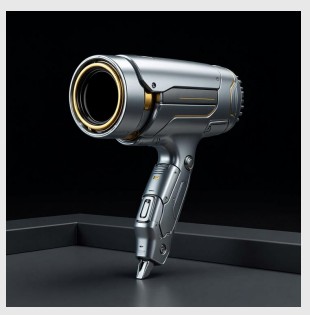

A stylish hairdryer placed on a vanity table. The design style of the hairdryer is inspired by the comb in the given image.

Figure 28: Multimodal instruction-based generation cases of DreamOmni2.

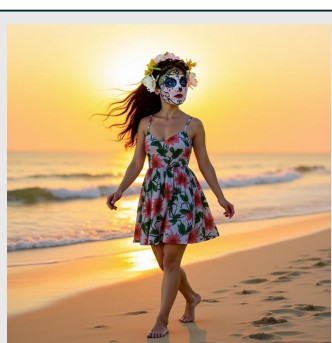

Generate a woman in a floral summer dress, walking barefoot along a beach at sunset. Her hair flows in the breeze, and she smiles softly as she watches the waves. Her makeup is the same as the woman in the given image. The background features a golden sun dipping below the horizon, casting warm hues over the calm ocean and sandy shore.

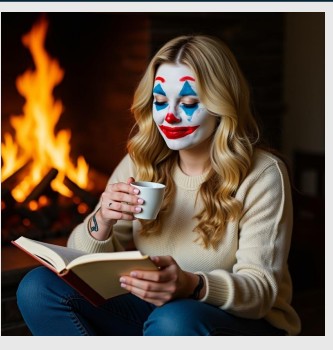

A woman in a cozy knitted sweater and denim jeans, sitting by a fireplace, sipping tea while reading a book. Her hair is styled in loose waves, and she has a calm, content expression. Her makeup is the same as the woman in the given image.

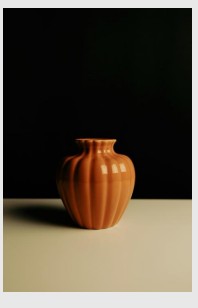

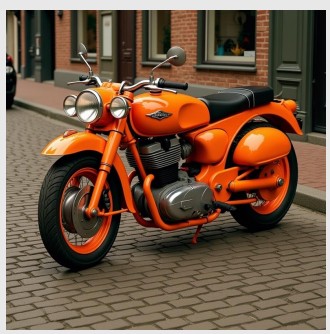

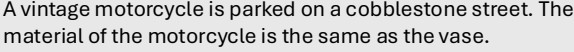

A vintage motorcycle is parked on a cobblestone street. The material of the motorcycle is the same as the vase.

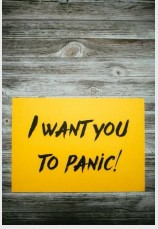

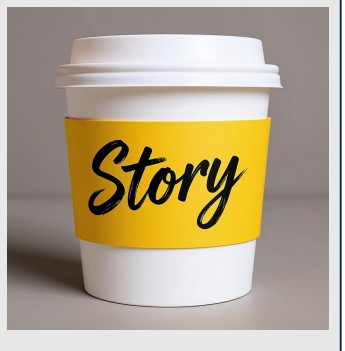

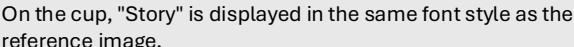

On the cup, "Story" is displayed in the same font style as the reference image.

Figure 29: Multimodal instruction-based generation cases of DreamOmni2.

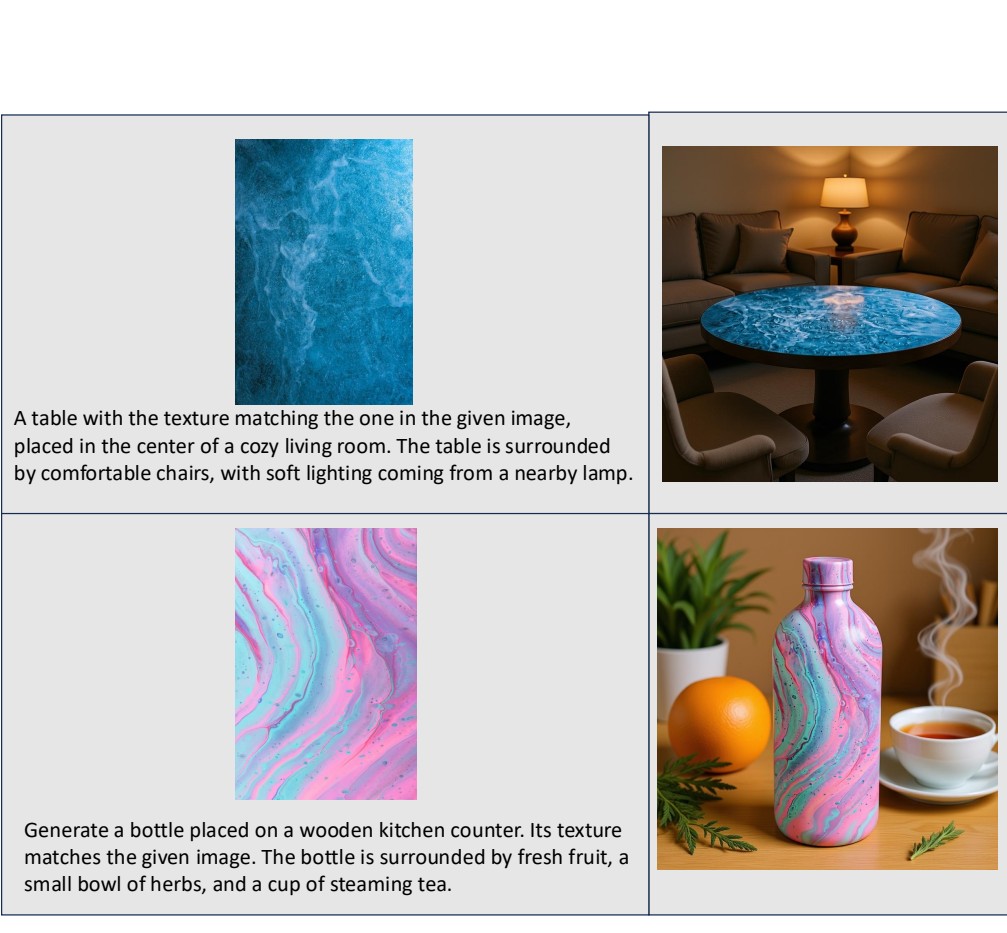

Figure 30: Multimodal instruction-based generation cases of DreamOmni2.

