# OpenReview forum: "DreamOmni2: Multimodal Instruction-based Editing and Generation"
_ICLR.cc/2026/Conference — ICLR 2026 Conference Withdrawn Submission_

### Official Review · Reviewer_Yy5h · 2025-10-30

**Soundness:** 3
**Presentation:** 3
**Contribution:** 3
**Rating:** 6
**Confidence:** 3

**Summary:**

The paper proposes DreamOmni2, a unified model framework designed for multimodal instruction-based image editing and generation. This framework allows users to provide both text and multiple images instructions, enhancing both concrete and abstract visual content creation. The approach leverages a comprehensive three-stage data pipeline and a unique joint training scheme for both a generation/editing model and a vision-language model (VLM).

**Strengths:**

1. The paper introduces two highly practical tasks — multimodal instruction-based editing and generation, significantly advancing the capabilities of image models. By handling both abstract and concrete concepts, DreamOmni2 is a step forward in bridging the gap between simple object manipulations and more complex conceptual image transformations.

2. The paper presents a detailed and innovative three-stage data synthesis pipeline that allows for the generation of high-quality training data for both the extraction and editing tasks. The feature mixing method is particularly compelling, as it avoids edge-blending issues in previous approaches, which contributes to higher quality output.

3. The incorporation of a Vision-Language Model (VLM) for joint training improves the model’s understanding of complex user instructions, making the system more adaptable to real-world use cases.

4. The DreamOmni2 benchmark is well-constructed, including both concrete object and abstract attribute tasks. This allows for meaningful performance evaluation and comparison with existing models, showing DreamOmni2’s superior capability in handling diverse editing and generation requests.

**Weaknesses:**

1. The three-stage data pipeline, while innovative, may be difficult to scale or implement outside the authors’ controlled setup. More details on how to adapt this pipeline for broader applications (e.g., other domains or industries) would help in assessing the practical value of DreamOmni2 beyond the experimental context.

2. Insufficient Evaluation Methodology: The paper primarily relies on VLM-based and human evaluations, which, while useful, lack more objective and rule-based evaluation methods to provide comprehensive assessment. Notably, the work lacks comparison and discussion with DreamBench++ (Peng et al., 2024), which addresses similar evaluation challenges by replacing DINO-based metrics with GPT-based evaluations that demonstrate better correlation with human judgment. Incorporating such established evaluation frameworks or developing comparable rule-based metrics would strengthen the credibility and reproducibility of the reported results, making the claims more verifiable beyond subjective assessments.

**Questions:**

How does the method ensure the diversity and representativeness of the synthetic data used for training, particularly for abstract attributes?

---

### Official Review · Reviewer_tA9m · 2025-10-31

**Soundness:** 3
**Presentation:** 1
**Contribution:** 3
**Rating:** 4
**Confidence:** 5

**Summary:**

The paper proposes a pipeline for constructing a multimodal instruction-based editing and generation dataset and introduces a VLM-trained model based on Kontext. The results appear to demonstrate the model's strong ability to understand multimodal instructions. However, the paper lacks a significant amount of necessary details, making it academically unsuitable for acceptance at this stage.

**Strengths:**

- The paper proposed a pipeline for constructing a multimodal instruction-based editing and generation dataset.
- The experimental results appear very promising.

**Weaknesses:**

The writing of the paper is excessively poor, lacking a substantial amount of detail, which makes it very difficult to follow. See Questions for details.

**Questions:**

- How was the extraction model trained, and what were its inputs and outputs? Figure 2 uses different image examples in Stage 2 and Stage 1, making it difficult to understand.

- How was the prompt in Stage 1 of Figure 2 constructed?

- How many training instances were ultimately constructed?

- How was the VLM integrated and trained with Kontext, and what was the data format?

- What are the computational details of the model's evaluation results, and how was successful editing determined?

- How was the human evaluation designed? For example, how many annotators were assigned to evaluate each instance, and how was the evaluation conducted?

---

### Official Review · Reviewer_hXfa · 2025-11-01

**Soundness:** 2
**Presentation:** 2
**Contribution:** 2
**Rating:** 4
**Confidence:** 4

**Summary:**

The paper targets unified multimodal instruction for both image editing and generation with multi-reference inputs, and it assembles all the essential ingredients: a clear problem setup, a data construction pipeline, a model built on a capable base, and evaluations that include both qualitative visualizations and quantitative scores. However, it lacks a clearly illustrated model architecture and more rigorously calibrated validation for the VLM-as-judge evaluation choice.

**Strengths:**

1. Clearly formulates a practical setting: multi-reference, multimodal instructions spanning both editing and generation, including abstract attribute transfer.
2. Provides both qualitative and quantitative evaluations, with side-by-side visual examples and automatic scoring to support claims.
3. Data pipeline is spelled out and reproducible.
4. Writing and organization are clean, making the overall contribution and experiment flow easy to track.

**Weaknesses:**

1. No model architecture diagram. The dataflow and where each tweak attaches to the backbone are not visualized, which is not clear.
2. Heavy reliance on VLM-based scoring plus limited human evaluation, with no calibration evidence (e.g., inter-rater agreement, correlation with humans, confidence intervals).
3. Lacks objective/quantitative metrics for controllability and fidelity.
4. Few comparisons on established single-image editing and T2I generation benchmarks, which are still necessary if the method truly subsumes the single-image case. Or maybe the author could emphasize and explain the differences from previous multi-image controllable image generation and editing methods.

**Questions:**

1. There is already a substantial of work that uses multimodal instructions rather than purely textual ones. In that context, does the abstract’s claim "relying only on language for instructions is a limitation" actually hold? Shouldn’t the paper more thoroughly compare with and discuss prior work based on multimodal instruction?
2. The paper lacks a model architecture diagram, which makes the method less intuitive to understand.
3. Is it reliable to evaluate solely with a VLM and human raters? Is there evidence that this is reliable? Why are there no additional image-side quantitative metrics?
4. If the method can perform multi-image editing, then it should also handle single-image editing and generation, therefore it ought to be evaluated and compared on prior single-image editing and T2I generation benchmarks as well.

---

### Note · Authors · 2025-11-16

I have read and agree with the venue's withdrawal policy on behalf of myself and my co-authors.